# Population genomics of apricots unravels domestication history and adaptive events

Alexis Groppi [1,2,19], Shuo Liu [3,4,19], Amandine Cornille [5,19], Stéphane Decroocq[3], Quynh Trang Bui[3], David Tricon[3], Corinne Cruaud[6,7], Sandrine Arribat[8], Caroline Belser [6,7], William Marande[8], Jérôme Salse[9], Cécile Huneau[9], Nathalie Rodde [8], Wassim Rhalloussi[8], Stéphane Cauet [8], Benjamin Istace [6,7], Erwan Denis [6,7], Sébastien Carrère[10], Jean-Marc Audergon[11], Guillaume Roch[11,12], Patrick Lambert[11], Tetyana Zhebentyayeva[13], Wei-Sheng Liu[4], Olivier Bouchez[14], Céline Lopez-Roques[14], Rémy-Félix Serre[14], Robert Debuchy [15], Joseph Tran[16], Patrick Wincker [6,7], Xilong Chen [5], Pierre Pétriacq [3], Aurélien Barre[1], Macha Nikolski [1,2], Jean-Marc Aury [6,7], Albert Glenn Abbott[17], Tatiana Giraud [18✉] & Véronique Decroocq [3✉]

Among crop fruit trees, the apricot (*Prunus armeniaca*) provides an excellent model to study divergence and adaptation processes. Here, we obtain nearly 600 Armeniaca apricot genomes and four high-quality assemblies anchored on genetic maps. Chinese and European apricots form two differentiated gene pools with high genetic diversity, resulting from independent domestication events from distinct wild Central Asian populations, and with subsequent gene flow. A relatively low proportion of the genome is affected by selection. Different genomic regions show footprints of selection in European and Chinese cultivated apricots, despite convergent phenotypic traits, with predicted functions in both groups involved in the perennial life cycle, fruit quality and disease resistance. Selection footprints appear more abundant in European apricots, with a hotspot on chromosome 4, while admixture is more pervasive in Chinese cultivated apricots. Our study provides clues to the biology of selected traits and targets for fruit tree research and breeding.

---

[1] Univ. Bordeaux, Centre de Bioinformatique de Bordeaux (CBiB), Bordeaux 33076, France. [2] Univ. Bordeaux, CNRS, IBGC, UMR 5095, Bordeaux 33077, France. [3] Univ. Bordeaux, INRAE, UMR 1332 Biologie du Fruit et Pathologie, 71 Av. E. Bourlaux, Villenave d'Ornon 33140, France. [4] Liaoning Institute of Pomology, Tiedong Street, Xiongyue, Bayuquan District, Yingkou City 115009 Liaoning, China. [5] Université Paris Saclay, INRAE, CNRS, AgroParisTech, UMR GQE-Le Moulon, Gif-sur-Yvette 91190, France. [6] Genoscope, Institut François Jacob, Commissariat à l'Energie Atomique (CEA), Université Paris-Saclay, 2 Rue Gaston Crémieux, Evry 91057, France. [7] Génomique Métabolique, Genoscope, Institut François Jacob, CEA, CNRS, Univ Evry, Université Paris-Saclay, 2 Rue Gaston Crémieux, Evry 91057, France. [8] French Plant Genomic Resource Center, INRAE-CNRGV, Castanet Tolosan, France. [9] INRAE/UBP UMR 1095 GDEC Genetique, Diversite et Ecophysiologie des Cereales, Laboratory PaleoEVO Paleogenomics & Evolution, 5 Chemin de Beaulieu, Clermont Ferrand 63100, France. [10] LIPME, Université de Toulouse, INRAE, CNRS, Castanet-Tolosan, France. [11] INRAE UR1052 GAFL, Domaine Saint Maurice, CS60094 Montfavet 84143, France. [12] CEP INNOVATION, 23 Rue Jean Baldassini, Lyon 69364 Cedex 07, France. [13] The Schatz Center for Tree Molecular Genetics, Department of Ecosystem Science and Management, The Pennsylvania State University, University Park 16802 PA, USA. [14] INRAE, US 1426, GeT-PlaGe, Genotoul, Castanet-Tolosan 31326, France. [15] Université Paris-Saclay, CEA, CNRS, Institute for Integrative Biology of the Cell (I2BC), Gif-sur-Yvette 91198, France. [16] EGFV, Bordeaux Sciences Agro, INRAE, Univ. Bordeaux, ISVV, Villenave d'Ornon 33882, France. [17] Forest Health Research and Education Center, University of Kentucky, Lexington, KY, USA. [18] Ecologie Systématique et Evolution, CNRS, Université Paris-Saclay AgroParisTech, Orsay 91400, France. [19] These authors contributed equally: Alexis Groppi, Shuo Liu, Amandine Cornille. ✉email: veronique.decroocq@inrae.fr; tatiana.giraud@u-psud.fr

Domestication involves recent and strong selection, leaving adaptation footprints in the genome that are easier to detect than those left by natural selection[1]. A number of genome-scan studies on adaptive evolution in domesticated annual plants such as maize and rice have led to the identification of candidate regions for important traits[2–5]. Furthermore, independent domestication events have occurred in some crops, fungi and animals with selection on the same traits, resulting in convergent adaptation[6,7], such as the loss of seed shattering, minimization of seed dormancy and increase in seed size and number in annual crops[8]. These independent domestication events having led to similar derived traits provide opportunities to address the question of whether such convergent adaptation occurs through changes in the same or different genomic regions. For example, the loss of seed shattering, minimization of seed dormancy and increase in seed size and number arose in different crops through different genomic changes in various species[8,9]; in contrast, the loss of seed shattering has the same genetic basis in sorghum, rice, maize and foxtail millet[10–12].

The impact of domestication on genomes has been mainly studied in annual crops and seldom investigated in fruit tree crops[13]. The long juvenile phases, large effective population sizes and high outcrossing rates often found in trees may have limited the loss of genetic diversity and the impact of selection in genomes compared to selfing annual plants[14]. In long-lived fruit trees, human selection nevertheless acted on reproductive traits (e.g., mating system and flowering time)[15], vegetative traits (e.g., reduction of the juvenile phase and graft compatibility)[14,16,17], on fruit traits (fleshy fruit, size, acidity, firmness, flavor)[18], as well as on response to biotic (fungi, bacteria, insects, and weeds) and abiotic stresses (drought, salt, and cold)[19,20]. However, compared with annual crops, in perennial plant species relatively little is known about the impacts on genome structure and function during adaptive trait evolution in response to human selection[13,21].

In temperate regions of the Northern and Southern hemispheres, apricots are cultivated for their fruits and flowers, and sometimes their kernels. They belong to the Armeniaca section of the family Rosaceae, subfamily Prunoideae. *Prunus armeniaca* L. refers to both the wild progenitor and the cultivated species (also called 'common apricot'). It is a deciduous tree grown for its edible fruits with an annual worldwide production of ~4.1 million tons (FAO, 2019). It is mostly cultivated in the Mediterranean region (Turkey as the largest producer, mainly of dried apricots), the Middle East, in the Caucasus, Central Asia (with Uzbekistan as the second largest producer) and China. Natural populations of *P. armeniaca* still occur, but only in Central Asia[22–24]. *Prunus mume*, a related species within the Armeniaca section (Siebold) Siebold & Zucc., is primarily cultivated for its flowers and secondly for its fruits, consumed as salted and smoked. The four other related species are *P. sibirica* L., *P. mandshurica* (Maxim.) Koehne, *P. holosericea* (also viewed as a variant of *P. armeniaca* and called *Prunus armeniaca* var. *holosericea* Batalin[25]) and *P. brigantina*; the first three are endemic in Eastern Asia (mostly China), while the more distant *P. brigantina* Vill. occurs in the French and Italian Alps[26–28]. All these species are diploid ($2n = 16$) with relatively small genome sizes (~220–230 Mbp), which, together with the availability of wild gene pools, make apricot an excellent system to study the domestication process in perennial tree crops.

The history of apricot domestication and the impact of adaptive trait evolution on the genome remain unclear. Based on morphological and botanical data, apricot was considered for a long time to have originated in China[29]. However, recent population genetics studies showed a closer relationship of European *P. armeniaca* apricots with wild Central Asian populations than

with Chinese apricots, suggesting the existence of multiple independent domestication events in Central Asia, Europe and China, although the populations-of-origin could not be identified[24,27,30]. European and Chinese cultivated apricots share similar specific crop features, such as fruit shape and size, as well as tree phenology, suggesting convergent adaptation during parallel domestication. However, the impacts of gene flow and selection during these domestication events have not been studied using high-quality apricot genome assemblies or taking into account heterozygosity and previous genetic maps with identified quantitative trait loci (https://www.rosaceae.org/search/qtl).

In this work, we produce four high-quality and chromosome-scale assemblies of *P. armeniaca*, *P. sibirica* and *P. mandshurica* species. We also sequence the genomes of 578 Armeniaca individuals (Supplementary Note 1, Supplementary Data 1). We reveal that the Chinese and European cultivated apricots result from independent domestication events from distinct wild populations. We find that a relatively small part of their genomes is affected by selection, as expected for perennial crops, and that different genomic regions are affected by selection in European and Chinese cultivated apricots despite convergent phenotypic traits. Selection footprints appear more abundant in European apricots, with a hotspot on chromosome 4, while admixture is more pervasive in Chinese cultivated apricots. In both cultivated groups, however, the genes affected by selection have predicted functions important to the perennial life cycle, fruit quality and disease resistance.

## Results

*Four high-quality genome assemblies of Armeniaca species.* We de novo sequenced the following four Armeniaca genomes, using both long-read and long-range technologies: *Prunus armeniaca* accession Marouch #14, *P. armeniaca* cv. Stella, accession CH320_5 sampled from the Chinese North-Western *P. sibirica* population (Fig. 1a), and accession CH264_4 from a Manchurian *P. mandshurica* population (Fig. 1a).

Two *P. armeniaca* genomes, Marouch #14 and Stella, were sequenced with the PacBio technology (Pacific Biosciences), with a genome coverage of respectively 73X and 60X (Supplementary Note 2) and assembled with FALCON[32] (Supplementary Figs. 1 and 2). To further improve these assemblies, we used optical maps to perform hybrid scaffolding and short reads[33] to perform gap-closing[34]. Because of their self-incompatibility, and thus expected higher rate of heterozygosity (Supplementary Fig. 3), *P. sibirica* and *P. mandshurica* were sequenced and assembled using different approaches. Both were sequenced using ONT (Oxford Nanopore Technologies), with a genome coverage of 113X and 139X, respectively. Raw reads were assembled and resulting contigs were ordered using optical maps (Bionano Genomics). Manual filtering during the integration of optical maps and subsequent allelic duplication removal helped resolve the heterozygosity-related issues in the assemblies (see Methods and Supplementary Note 3).

The Marouch and Stella assemblies were then organized into eight pseudo-chromosomes using a set of 458 previously published molecular markers, whereas the chromosomal organization of CH320-5 and CH264-4 assemblies were obtained by comparison with *P. armeniaca* pseudo-chromosomes (Supplementary Note 3). Baseline genome sequencing, RNA sequencing, analyses and metadata for the four de novo assembled genomes are summarized in Table 1, Supplementary Notes 3 and 4, and Supplementary Data 2–4. We found high synteny between our assemblies and the two available apricot genome assemblies of similar high quality[35,36], with, however, rearrangements around centromeres (Supplementary Note 4; Supplementary Data 5,

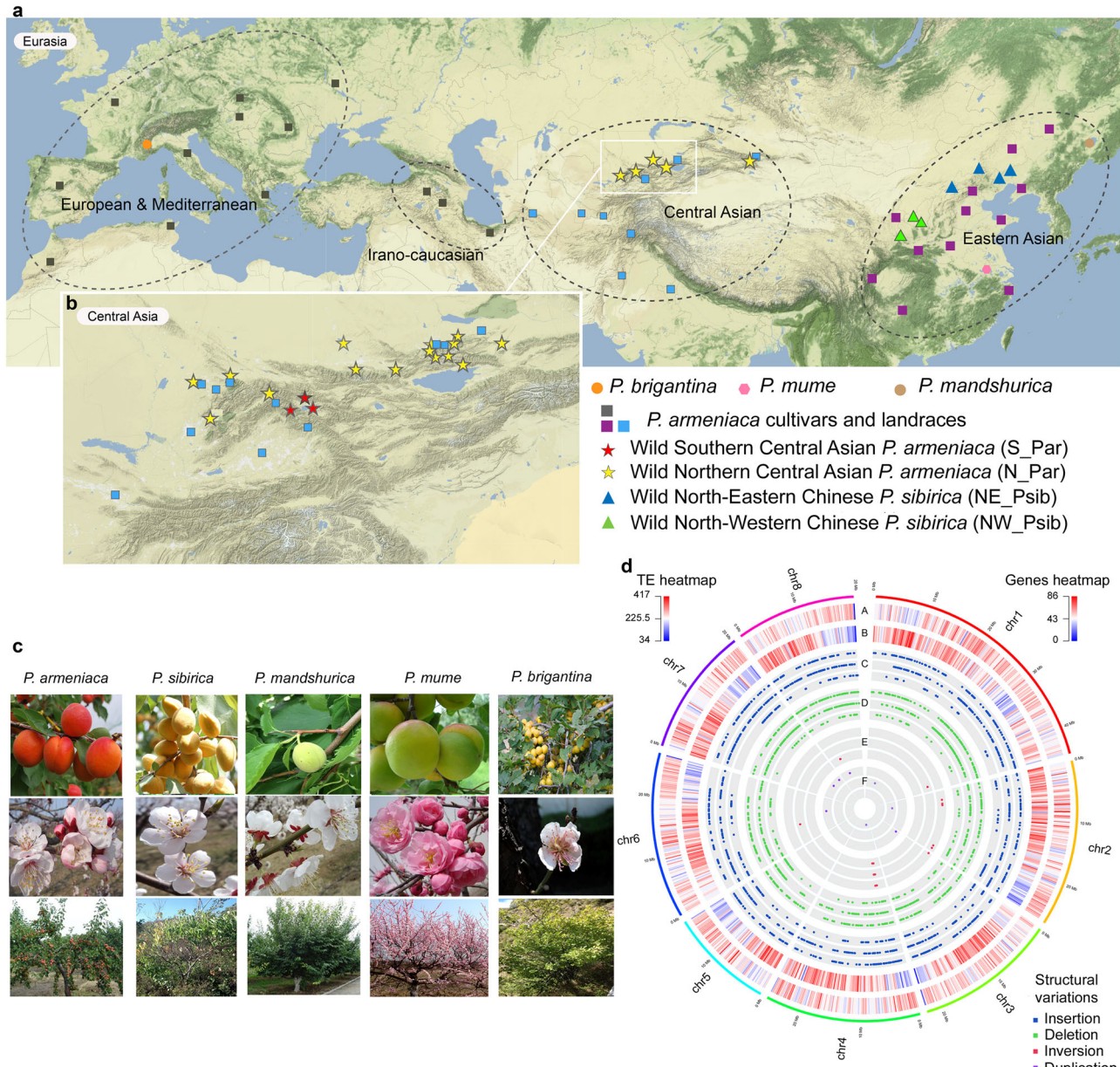

**Fig. 1 Geographical distribution and features of Armeniaca species. a** Map of species distribution and of plant material used in this study (Supplementary Data 1). The European and Irano-Caucasian cultivated apricots include 39 modern cultivars from North America, South Africa and New Zealand that are not represented on this map. Orange circles: *P. brigantina*, pink circles: *P. mume*, beige circles: *P. mandshurica*; rectangles: *P. armeniaca* cultivars and landraces (European in grey, Chinese in purple, Central Asian in blue); red stars: wild Southern Central Asian *P. armeniaca* (S_Par); yellow stars: wild Northern Central Asian *P. armeniaca* (N_Par); blue triangles: wild Northern Eastern Chinese *P. sibirica* (NE_Psib); green triangles: wild Western Chinese *P. sibirica* (NW_Psib). **b** Close-up of the Central Asian *Prunus armeniaca* (Par) natural population. Northern *P. armeniaca* populations (N_Par) are located over the Zailiysky Mountains while Southern *P. armeniaca* (S_Par) populations lay over the foothills of the Ferghana valley. **c** Pictures of the species analysed. **d** Circos plot showing features of the Marouch #14 *Prunus armeniaca* genome. Outermost to innermost tracks show the gene density heat map in 250 kb bins (**a**), transposable element (TE, **b**) density in 250 kb bins. The red color indicates higher density while the blue color indicates lower density. **c–f** represent structural variations (insertions in blue, deletions in green, inversions in red, and duplications in purple). For each track from (**c**)–(**f**), the outer layer corresponds to *P. armeniaca* cv. Stella, the middle layer to the *P. sibirica* CH320.5 and the inner layer to the *P. mandshurica* CH264.4 line. Maps are licensed as Creative commons for Stamen design. Pictures by S. Liu, S. Decroocq or licensed as Creative commons (*P. brigantina* flower). The picture of cv. Sefora apricot (*P. armeniaca*) fruits was kindly provided by CEP INNOVATION. Source data underlying Fig. 1d are provided as a Source Data file and in Supplementary Data 8.

Supplementary Figs. 2–5). Molecular markers (SSR and SNP) have been used to check some structural variations and to align physical and genetic maps (Supplementary Data 6).

*Characteristics of the Armeniaca genomes and patterns of structural variation.* The heterozygosity rate estimated from the

corrected reads (Supplementary Note 4) indicated that the least heterozygous assembled genome was the apricot Marouch #14 accession (0.2% of heterozygosity), then cv. Stella (0.37%), Manchurian CH264_4 (0.82%) and Siberian CH320_5 (0.95%) (Supplementary Fig. 3). These differences are consistent with the self-incompatible reproductive system of the wild Armeniaca

**Table 1 Statistics for the genome assemblies of Armeniaca species.**

|  | P. armeniaca | | P. sibirica | P. mandshurica |
|  | Marouch #14 | cv. Stella | CH320-5 | CH264-4 |
|---|---|---|---|---|
| Final assembly length | 203.93 Mb | 212.06 Mb | 259.43 Mb | 223.66 Mb |
| Number of scaffolds | 8 | 12[a] | 8 | 8 |
| Scaffold N50 (L50) | 25.15 Mb (4) | 25.5 Mb (4) | 33.8 Mb (4) | 29.44 Mb (4) |
| Scaffold N90 (L90) | 20.01 Mb (7) | 20.01 Mb (7) | 26.17 Mb (7) | 22.24 Mb (7) |
| Maximum length of scaffold | 44.41 Mb | 43.52 Mb | 49.18 Mb | 39.9 Mb |
| Number of contigs | 302 | 391 | 517 | 286 |
| Contig N50 (L50) | 1.8 Mb (31) | 1.3 Mb (40) | 1.7 Mb (38) | 3.2 Mb (20) |
| Percentage of Ns[b] | 0.09% | 0.13% | 7.15% | 1.42% |
| Heterozygosity rate[c] | 0.23% | 0.37% | 0.95% | 0.82% |
| Estimated genome size (bp) | 236.36 Mb | 230.26 Mb | 225.33 Mb | 231.77 Mb |
| Number of genes | 37,521 | 38,237 | 43,741 | 39,021 |
| Number of predicted proteins | 40,067 | 40,960 | 46,196 | 41,386 |
| Mean coding sequence (CDS) length (bp) | 1255 | 1250 | 1163 | 1235 |
| Mean exons per CDS | 4.8 | 5 | 4.5 | 4.8 |
| BUSCO scores[d] | 97.30% | 97.80% | 98.20% | 94.10% |

Heterozygosity and genome size were estimated by GenomeScope 2.0 (Supplementary Note 4).
[a]Eight pseudo-chromosomes and four unplaced scaffolds.
[b]Percentage of base pairs with uncertain sequence.
[c]1% heterozygosity rate corresponds to 1 SNP per 100 bp.
[d]Percentage of genes present in the genome compared to the BUSCO list.

Manchurian CH264_4 and Siberian CH320_5 trees and Chinese cultivated apricots[37], while 51–58% of the modern and traditional European apricots are self-compatible, as is Marouch #14[38,39].

The Marouch #14 apricot genome contains 37,521 predicted genes. Compared to the embryophyta_odb10 BUSCO set of orthologs, 97.30% of the predicted genes are full length, and only 2.1% are missing (Table 1; Supplementary Data 4; Supplementary Fig. 4). A total of 37.48% of the predicted open reading frames were identified as transposable elements (Supplementary Data 7). Based on thirteen P. armeniaca RNAseq datasets (Supplementary Data 2), we annotated between 40,067 and 46,196 proteins depending on the assembled genome (Table 1; Supplementary Note 5; Supplementary Fig. 6).

The number and class of transposable elements (TEs), as well as their relative abundance, showed considerable variation among the four genome assemblies (Supplementary Note 5; Supplementary Data 7; Supplementary Figs. 7 and 8). The most common class of TEs found in Armeniaca genomes corresponded to LTR (long terminal repeat) retrotransposons (Supplementary Fig. 8). We found a higher synteny between P. armeniaca Marouch #14 and cv. Stella (Supplementary Fig. 9) and to a lesser extent between Marouch #14 and Siberian CH320_5, while the P. mandshurica CH264_4 accession showed more re-arrangements when compared to the other apricot genomes (Supplementary Fig. 9). We observed few large structural variations between Marouch #14 and Stella or between Marouch #14 and CH320_5 or CH264_4 (Fig. 1d) (Supplementary Note 6; Supplementary Fig. 10 and 11; Supplementary Data 8–9). The structural variants were mostly insertions/deletions and ranged in size from 501 bp to 4.1 Mb, with a majority of variants smaller than 10 kb (Supplementary Fig. 12; Fig. 1d). In particular, an inversion of ca. 600 Kb was detected in the P. armeniaca Marouch #14 genome when compared to the three other genomes assembled in this study, P. armeniaca cv. Stella, Siberian CH320_5 and the P. mandshurica CH264_4 (Supplementary Data 8; Supplementary Fig. 10; Fig. 1d). This large inversion, validated by PCR (Supplementary Fig. 10), is located at the edge of chromosome 4 (approximately position 3.65 Mbp) and likely corresponds to a recent structural rearrangement as it is present only in the Marouch #14 genome. From a breeder's perspective, such information will be important when Marouch #14 is used as a reference genome for read mapping and when the Marouch #14 individual is used as a parent in crosses.

*Reconstruction of Armeniaca phylogeny.* A genome-wide analysis of fourfold degenerated (neutral) polymorphism of diploid Rosaceae species, together with three more distantly related species with known divergence times (i.e., between *Populus trichocarpa* and *Arabidopsis thaliana* or *Fragaria vesca* and *Rosa chinensis*[40,41]), estimated the divergence between Armeniaca and Amygdalus lineages >7 Mya (million years ago) (Supplementary Note 7, Supplementary Data 10, Fig. 2a and Supplementary Fig. 13), which is much later than previously suggested[42]. The phylogeny placed *P. mume*[43] as the first diverging lineage within the Armeniaca section (4 Mya); the *P. brigantina* lineage actually diverged first[27] but could not be incorporated in our phylogeny because its genome has not been assembled yet.

*Chromosome structural evolution in the Armeniaca clade.* In order to assess the chromosome structural evolution within the Rosaceae family, we reconstructed ancestral genomes[44] based on available Armeniaca genomes (*P. armeniaca* cv. Stella and Marouch #14, *P. sibirica* CH320_5, *P. mandshurica* CH264_4, *P. mume*) together with other public Rosaceae genomes (Fig. 2b) using grape as an outgroup (Supplementary Note 7). Conserved gene colocations among the eleven investigated genomes validated the previously published ancestral Rosaceae genome reconstruction into nine proto-chromosomes (Fig. 2b, Supplementary Fig. 14)[45]. The reconstructed Prunoideae ancestral genome with eight proto-chromosomes derived from the ancestral Rosaceae genome through two chromosome fissions and four fusions; the chromosome structure of the Siberian CH320_5 genome was the most similar to the inferred ancestral Rosaceae chromosomal arrangement (Fig. 2b). Our genome sequence-based chromosomal evolution study unraveled the Rosaceae karyotype history and identified shared orthologs in the apricot genomes (8,848 genes, Supplementary Data 10 and 11; Fig. 2c), that can be used for translational research among the investigated species to accelerate the dissection of conserved agronomic traits.

*Phylogenetic analysis of the Armeniaca chloroplast genomes.* Short-read sequencing data of 578 Armeniaca accessions (this

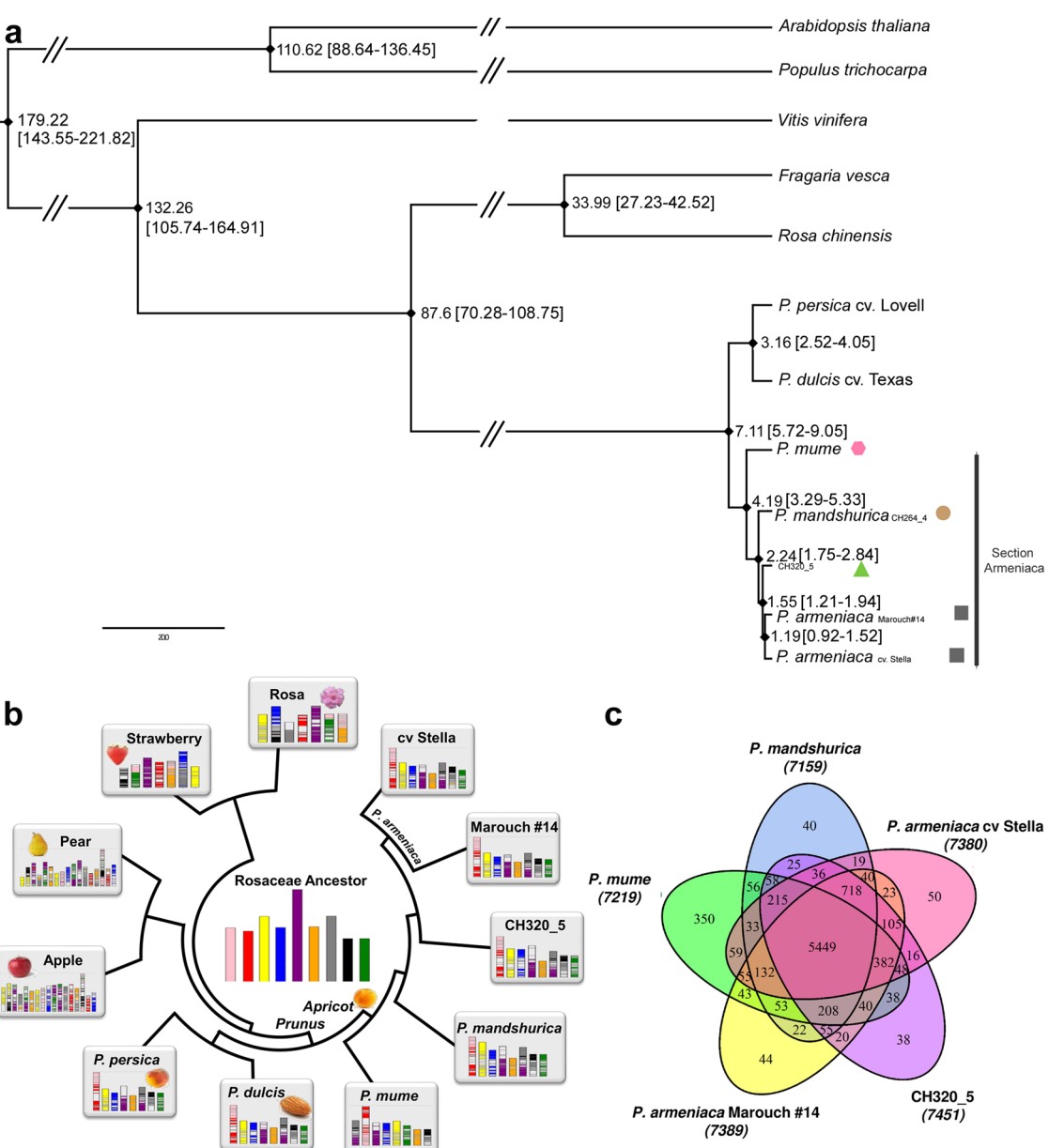

**Fig. 2 Reconstruction of Armeniaca phylogeny and chromosome structural evolution. a** Species tree. The phylogenetic tree was constructed on the basis of neutrally evolving sites from 298 shared single-copy orthologs. The values on the branch (in Mya) are the times of divergence estimated with BEAST and in brackets the confidence intervals. Pink circle: *P. mume*, beige circle: *P. mandshurica*; green triangle: *P. sibirica* CH320_5, grey rectangles: European *P. armeniaca* cultivars. **b** Chromosome structural evolution within Rosaceae. The modern Rosaceae genomes are illustrated with different (arbitrary) colors reflecting the origin from the nine chromosomes (center) of the inferred ancestral Rosaceae karyotype (ARK). **c** Numbers of ancestral Rosaceae genes conserved in the five modern apricot genomes shown in a Venn diagram, with arbitrary colors to better see the different groups. Source data are provided as a Source Data file and in Supplementary Data 10.

study; Supplementary Data 1), together with 15 available *P. mume* genomes[43], were used for reference-based reconstruction of chloroplast genomes (cpDNA, Supplementary Note 8). For phylogenetic inferences, we selected 2-4 chloroplast genomes per species, representing the cpDNA diversity of wild and cultivated *P. armeniaca*, *P. sibirica*, *P. mume* and *P. brigantina* populations. The cpDNA assembly of *Prunus padus* L. (KP760072) was included as an outgroup. The haplotype network of chloroplast genomes closely mirrored the pattern observed on the maximum likelihood tree (Supplementary Note 8; Fig. 3 and Supplementary Fig. 15). Three closely related cpDNA haplotypes were found in most *P. armeniaca* individuals (A1, A2, A3, in both wild and cultivated groups; Fig. 3). While the three haplotypes A1, A2, and

A3 were present in Central Asian and Chinese *P. armeniaca* populations, European cultivated apricots displayed either the A1 or the A2 haplotype. Some of the *P. sibirica* chloroplast genomes were indistinguishable from those found in *P. armeniaca*, harboring the A1, A2 or A3 haplotypes, while other *P. sibirica* chloroplast genomes were instead resolved as a sister group to *P. brigantina* with maximum support (Supplementary Fig. 15); the finding of intermingled *P. armeniaca* and *P. sibirica* chloroplast genomes suggests hybridization or misclassification.

*Evolutionary history of wild and cultivated apricots.* To investigate the genetic diversity and evolutionary history of Armeniaca lineages, we analysed the genomes (ca 21x coverage) of 564

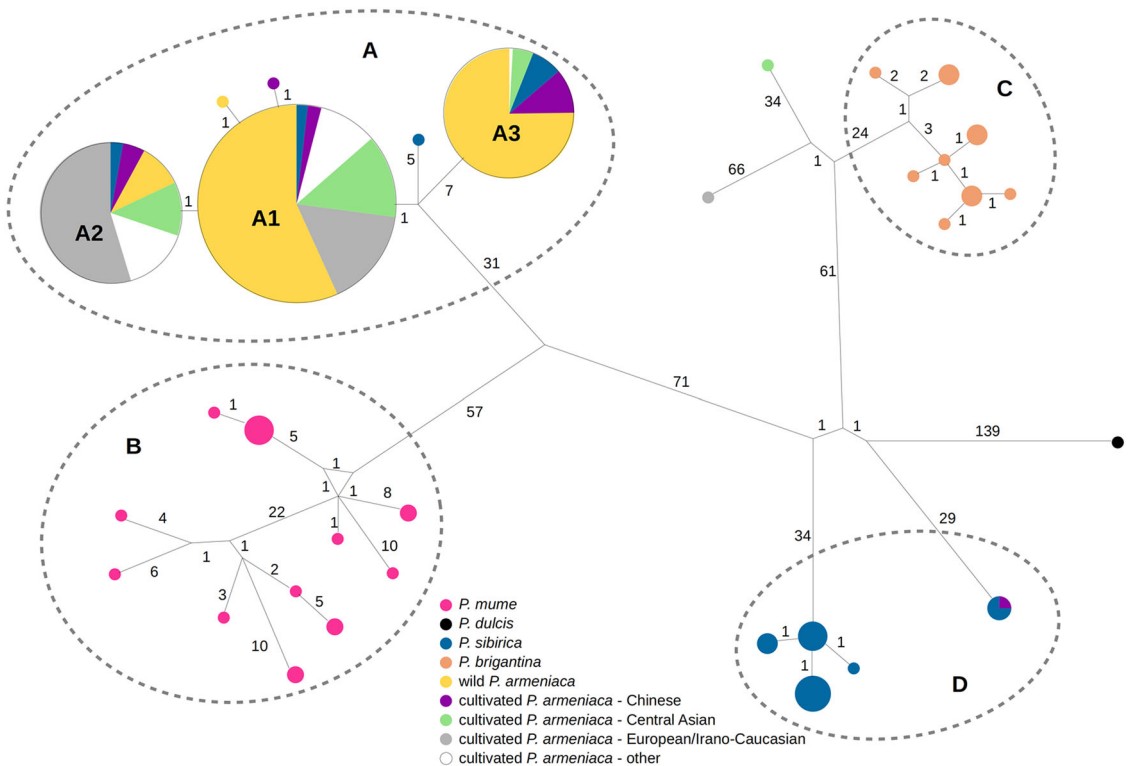

**Fig. 3 Median-joining network showing the genetic relationships among Armeniaca chloroplastic haplotypes.** The four main clusters are indicated as (**a**)–(**d**). Numbers under the lines indicate the number of substitutions separating two haplotypes. The taxa are colored according to the nuclear DNA-based groups (Fig. 1a and 1b). Pink: *P. mume*, black: *P. dulcis*, dark blue: wild Northern Eastern Chinese *P. sibirica* (NE_Psib); green: wild Western Chinese *P. sibirica* (NW_Psib), orange: *P. brigantina*, yellow: wild *P. armeniaca* (S_Par and N_Par); purple: cultivated Chinese apricots, light blue: cultivated Central Asian apricots, grey: cultivated European/Irano-Caucasian apricots, white: other cultivated apricots. The size of the circles indicates sample size. The two samples closer to *P. brigantina* correspond to interspecific plum x apricot individuals (A3865 and US196, Supplementary Data 1). Source data are provided as a Source Data file.

apricot and apricot-related species, including 256 wild *P. armeniaca* trees from Central Asian natural populations (Fig. 1b), 43 wild *P. sibirica* trees from eight Chinese natural populations, one *P. mandshurica* (Fig. 1a), and 264 cultivated *P. armeniaca* apricot accessions, comprising 27 Chinese, 166 European-Irano Caucasian and 71 Central Asian apricot cultivars (Supplementary Data 1 and Fig. 1; Supplementary Notes 1 and 9, Supplementary Fig. 16). We also used previously published genomes of *P. mume* ($N = 348$)[31]. Fourteen accessions of *P. brigantina* were used as outgroups[27]. SNPs were called using GATK best practices for this collection of 926 individuals (Supplementary Note 9). For population genetic structure inferences, genetic diversity and differentiation analyses, only Armeniaca species were retained (Supplementary Data 1). After SNP calling (Supplementary Note 9), a set of 15,111,266 SNPs was used in the following population-based genomic analyses.

We estimated linkage disequilibrium (LD) using the squared correlation coefficient ($r^2$) between pairs of SNPs over a 300 Kb physical distance in each of six sets: 348 *P. mume* samples and 555 other Armeniaca samples (i.e., mainly *P. armeniaca* and *P. sibirica* accessions, corresponding to the European and Chinese cultivars as well as the wild Central Asian and Chinese apricots) (Supplementary Note 9 and Supplementary Fig. 17). *Prunus mume* showed the highest LD level, likely because the 348 *P. mume* are mostly cultivated accessions (Supplementary Fig. 17a), and therefore not a panmictic population. LD was also higher in European cultivars than in the wild *P. armeniaca* populations (Supplementary Fig. 17b). Linkage disequilibrium nevertheless

decayed very quickly in all groups within a few hundred base pairs, along the eight chromosomes (Supplementary Fig. 17b), as previously shown[33].

*Armeniaca population subdivision and admixture.* We analysed the population subdivision and gene flow among apricot populations (Supplementary Notes 10 and 11). In the principal component analysis (PCA) based on a set of filtered 95,686 SNPs (MAF > 0.05 and LD pruned, Supplementary Note 11), most *P. armeniaca* and *P. sibirica* individuals formed a single cluster spread along the PCA axis 2, differentiated from the 14 *P. brigantina* and the 348 *P. mume* individuals along the PCA axis 1 (Fig. 4a). Excluding the two most differentiated species (*P. brigantina* and *P. mume*) and the single *P. mandshurica* individual, a second PCA (Fig. 4b) showed a clearer genetic differentiation between the *P. armeniaca* and *P. sibirica* populations, except for the North Western *P. sibirica* individuals (NW_Psib) that grouped with Chinese cultivated apricots. Excluding the North Eastern *P. sibirica* from a third PCA, the Central Asian individuals fell between the two well differentiated clusters of *P. armeniaca*, the European cultivated apricots (in grey, Fig. 4c) and the wild Central Asian *P. armeniaca* populations (red and yellow, Fig. 4c) while the North-Western *P. sibirica* individuals still grouped with Chinese apricots. This result indicates that the North Western *P. sibirica* individuals had been mis-assigned to the *P. sibirica* species while they belonged to *P. armeniaca*.

We also ran fastSTRUCTURE on the entire Armeniaca dataset ($n = 917$) from $K = 2$ to $K = 12$, revealing population subdivision

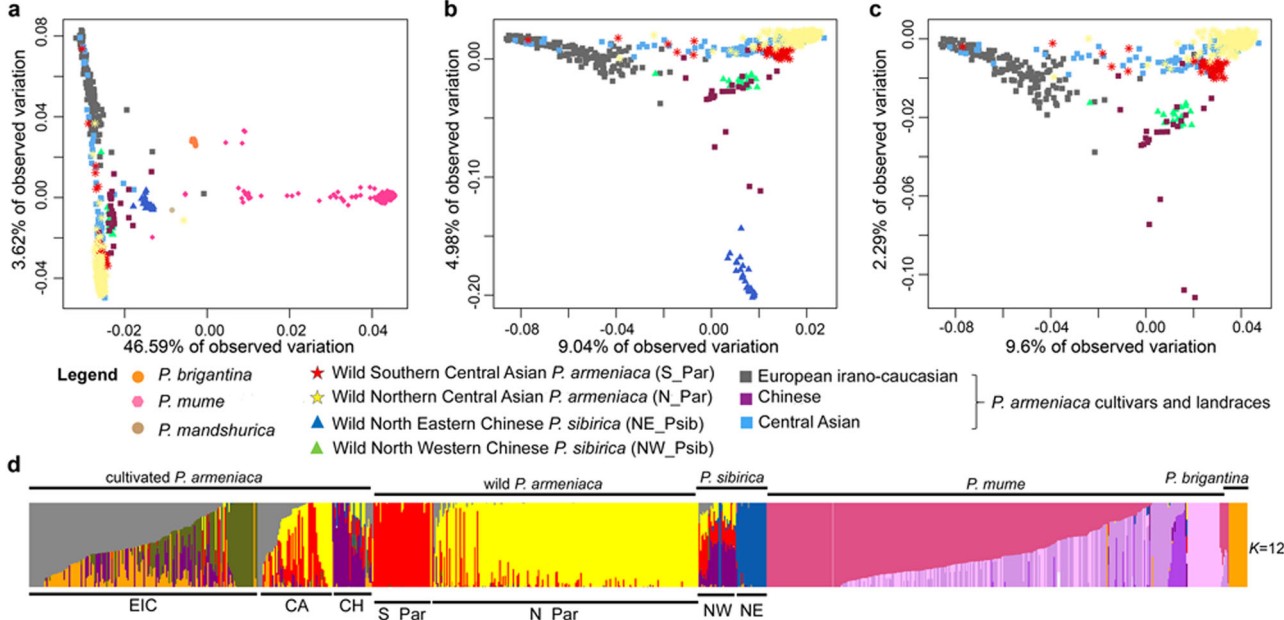

**Fig. 4 Population structure and variation in Armeniaca wild and cultivated apricots. a** Principal component analysis (PCA) from a set of 95,686 SNPs; the first PCA component explained 46.59% of total variation, the second component only 3.62%. The colors and signs correspond to the ones depicted in Fig. 1a-b. **b** PCA after removal of *P. mume*, *P. brigantina* and *P. mandshurica* individuals. Orange circles: *P. brigantina*, pink circles: *P. mume*, beige circles: *P. mandshurica*; rectangles: *P. armeniaca* cultivars and landraces (European in grey, Chinese in purple, Central Asian in blue); red stars: wild Southern Central Asian *P. armeniaca* (S_Par); yellow stars: wild Northern Central Asian *P. armeniaca* (N_Par); blue triangles: wild Northern Eastern Chinese *P. sibirica* (NE_Psib); green triangles: wild Western Chinese *P. sibirica* (NW_Psib). **c** PCA after removal of *P. sibirica* accessions from the NE_Psib populations. **d** fastSTRUCTURE barplot of the 917 Armeniaca individuals at *K* = 12 and 95,686 SNPs. EIC, European Irano Caucasian; CA, Central Asian; CH, Chinese; S_Par, Southern Central Asian *P. armeniaca*; N_Par, Northern Central Asian *P. armeniaca*; NW, North_Western *P. sibirica*; NE, North_Eastern *P. sibirica*. Source data are provided as a Source Data file.

consistent with the PCA results, with *P. mume* being highly differentiated from the rest of the Armeniaca samples (Fig. 4d and Supplementary Fig. 18). The Armeniaca samples, outside of *P. mume*, were further subdivided into seven groups corresponding to the cultivated and wild clusters of *P. armeniaca*, whereas the Chinese cultivated accessions and the North-Western wild *P. sibirica* formed a highly admixed group, differentiated from *P. sibirica* North Eastern populations and from the other *P. armeniaca* (Fig. 4d).

ABBA-BABA tests and fastSTRUCTURE analyses (Supplementary Notes 10 and 11, Supplementary Data 12; Supplementary Figs. 18–21) indicated that all Armeniaca genetic clusters, except *P. brigantina* and North Eastern *P. sibirica*, showed high levels of genetic admixture. This was especially true for the Central Asian and Chinese cultivated apricots and *P. mume* accessions, the first two ones showing admixture with the wild *P. armeniaca* and the last one with other Armeniaca and *Prunus* species[31]. Gene flow from North Eastern *P. sibirica* (NE_Psib) was only detected toward the Chinese cultivated apricots (Supplementary Note 11).

*Demographic inferences provide insight into the origin of cultivated apricots.* We reconstructed the evolutionary history of the various identified apricot gene pools (Supplementary Notes 11 and 12, Supplementary Figs. 27–31) using random forest approximate Bayesian computation (ABC-RF). We filtered out individuals identified as clonemates and siblings of other individuals (Supplementary Data 13), as well as admixed individuals (Supplementary Figs. 19–24), which included the Central Asian cultivated apricots (Fig. 4d, CA). We re-ran fastSTRUCTURE with this pruned dataset, and also removed genetic groups with recent admixture footprints, which included the mis-classified *P.*

*sibirica* accessions from the W3 cluster (Fig. 5a; Supplementary Note 12 and Supplementary Figs. 25–27). We indeed wanted to test whether there had been more ancient gene flow among the gene pools identified, without the signal being blurred by recent admixture that can be directly seen on barplots. Since *P. brigantina* is highly differentiated genetically from the other Armeniaca species and has a narrow geographic range, endemic to the Alps (Supplementary Note 12), it is an unlikely progenitor of cultivated apricots. The *P. brigantina* samples were thus removed for demographic inferences. We retained for demographic inferences 163 individuals belonging to six populations (Supplementary Data 14), defined as follows: European and Chinese cultivated apricots (C1 and CHN), wild *P. armeniaca* from Northern and Southern Central Asia (W1 and W2), wild *P. sibirica* (W4) and *P. mume* (Fig. 5a-b, and Supplementary Note 12). We kept 9613 synonymous unlinked SNPs common to the six populations.

ABC-RF supported an evolutionary history of wild apricots with gene flow among diverging lineages, with successive divergence of *P. mume* and the wild *P. sibirica* lineages (W4), and then of the wild Southern (W1) and Northern (W2) Central Asian *P. armeniaca* populations from the wild *P. sibirica* lineage (Fig. 5c; Supplementary Figs. 28–30 and Supplementary Data 15–18). ABC-RF inferences further supported the occurrence of gene flow during apricot domestication and independent domestication events having led to the two cultivated populations: Chinese cultivated apricots diverged from the wild Southern Central Asian *P. armeniaca* population (W1) ca. 2,900 ya while the European cultivated apricots diverged from the wild Northern Central Asian *P. armeniaca* populations ca. 2,250 ya (Fig. 5c-d; Supplementary Note 12 and Supplementary Data 16).

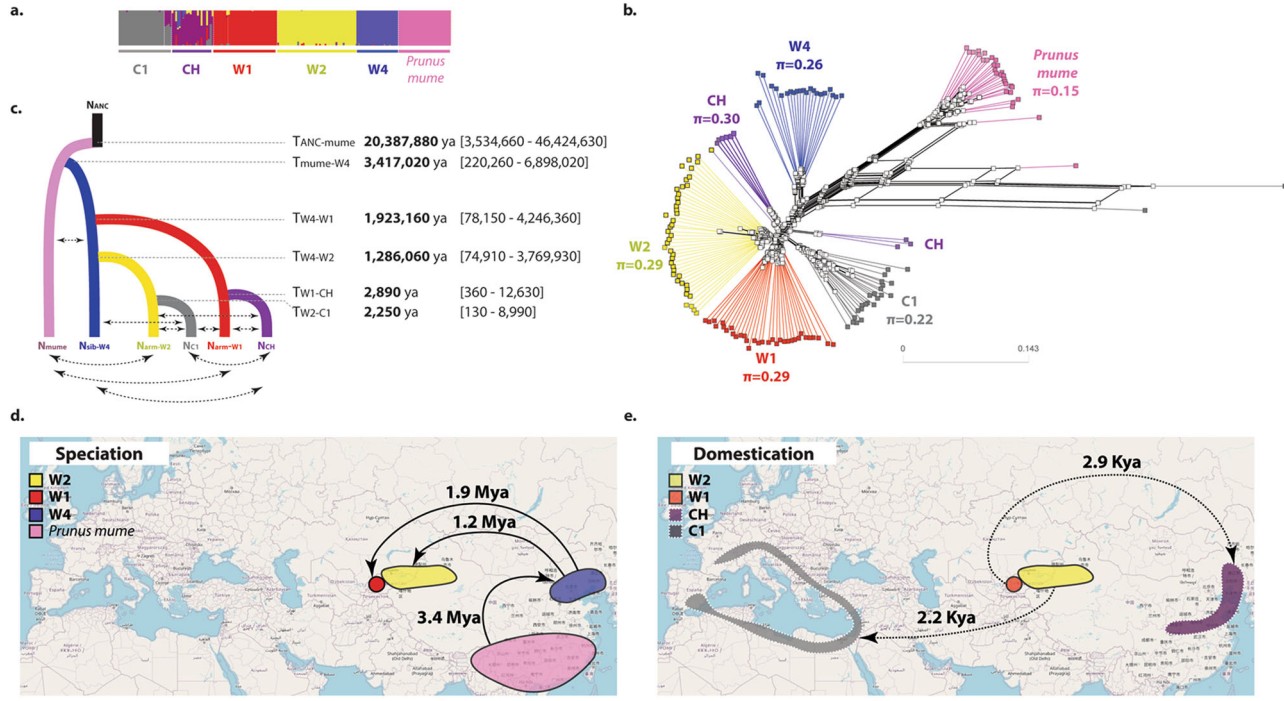

**Fig. 5 Inference of independent domestication events of apricot, with divergence in the face of gene flow, using random-forest approximate Bayesian computation combined with coalescent-based simulations. a** fastSTRUCTURE barplot for the pruned dataset used for random-forest approximate Bayesian computation (ABC-RF) computations, excluding *Prunus brigantina*, clonemates and siblings, as well as recently admixed individuals. **b** Splitstree for the pruned dataset used for ABC-RF computations, branches being colored according to the clusters identified with fastSTRUCTURE. Values under population labels are the average number of nucleotide differences between genotypes ($\pi$). **c** Most likely scenario of apricot domestication inferred from ABC-RF. Parameter estimates are shown, with their 95% confidence interval in brackets. Arrows represent migration between two populations. Associated maps depicting the speciation (**d**) and domestication (**e**) histories of apricots, with the approximate periods of time, drawn from ABC inferences. For all panels: W4 in blue: wild *Prunus. sibirica*; W1 in red and W2 in yellow: wild Southern and Northern Central Asian *P. Armeniaca*, C1 in grey and CH in purple: European and Chinese cultivated *P. armeniaca*, respectively, and *P. mume* in pink. Population names correspond to the ones detected with fastSTRUCTURE. Maps are licensed as Creative Commons. Source data are provided as a Source Data file.

*Evidence for post-domestication selection specific to Chinese and European apricot populations.* We looked for signatures of positive selection in the genomes of the two cultivated populations, the European cultivars originating from Northern Central Asian wild apricots, and the Chinese cultivars originating from Southern Central Asian populations. Most tests for detecting selection footprints are based on allelic frequencies, while admixture biases allelic frequencies. For selective sweep detection, we therefore used 50 non-admixed European cultivars with their two most-closely related wild Central Asian *P. armeniaca* populations, as inferred above in ABC-RF simulations (i.e., 33 W1 and 43 W2 accessions, respectively), and 10 non-admixed Chinese landraces with the wild *P. armeniaca* W1 populations (Supplementary Note 13; Supplementary Data 14).

*Genomic signatures of selection in cultivated apricot genomes.* A selective sweep results from selection acting on a locus, making the beneficial allele rise in frequency, leading to one abundant allele (the selected variant), an excess of rare alleles and increased LD around the selected locus. For detecting positive selection, we therefore used the composite-likelihood ratio test (CLR) corrected for demography history (Supplementary Fig. 31) and the Tajima's D, that detects an excess of rare alleles in the site-frequency spectrum (SFS) and we looked for regions of increased LD. We also used the McDonald-Kreitman test (MKT), that detects more frequent non-synonymous substitutions than expected under neutral evolution and we compared differentiation between cultivated populations and their genetically closest wild population through the population differentiation-based tests ($F_{ST}$ and $D_{XY}$)

to detect genomic regions more differentiated than genome-wide expectations (Supplementary Note 13, Supplementary Data 19 and 20).

Composite likelihood ratio (CLR) tests identified 856 and 450 selective sweep regions in the genomes of cultivated European and Chinese apricots, respectively (0.42% and 0.22% of the genome affected, respectively; Supplementary Data 21). The selective sweep regions did not overlap at all between the European and Chinese cultivated populations, suggesting the lack of parallel selection on the same loci despite convergent phenotypic traits (Supplementary Fig. 32). When taking as threshold the top 0.5% of CLR scores for European apricots, more than half of the selective sweeps detected (54 in total) were located in the middle of chromosome 4 (from 7 Mbp to 18 Mbp), indicating a potential hotspot of human selection targets (Fig. 6a-b) (Supplementary Note 14). In Chinese apricots, one third of the selective sweeps mapped on chromosome 1 and no particular enrichment was observed for chromosome 4 (Fig. 6c-d). We examined overlaps between known QTLs (quantitative trait loci) identified by GWAS (genome-wide association studies) or linkage mapping and the genomic regions with footprints of selection identified by the above tests. The apricot linkage group 4 was significantly enriched in selective sweeps associated with QTLs in its center region (from 7 to 18 Mbp coordinates), confirming a putative "hotspot" for important phenology traits (bloom and fruit maturity date) and for fruit quality traits (ripening, firmness, aroma) in European cultivated apricots, as previously suggested in cherry and peach[46,47] (Supplementary Note 14).

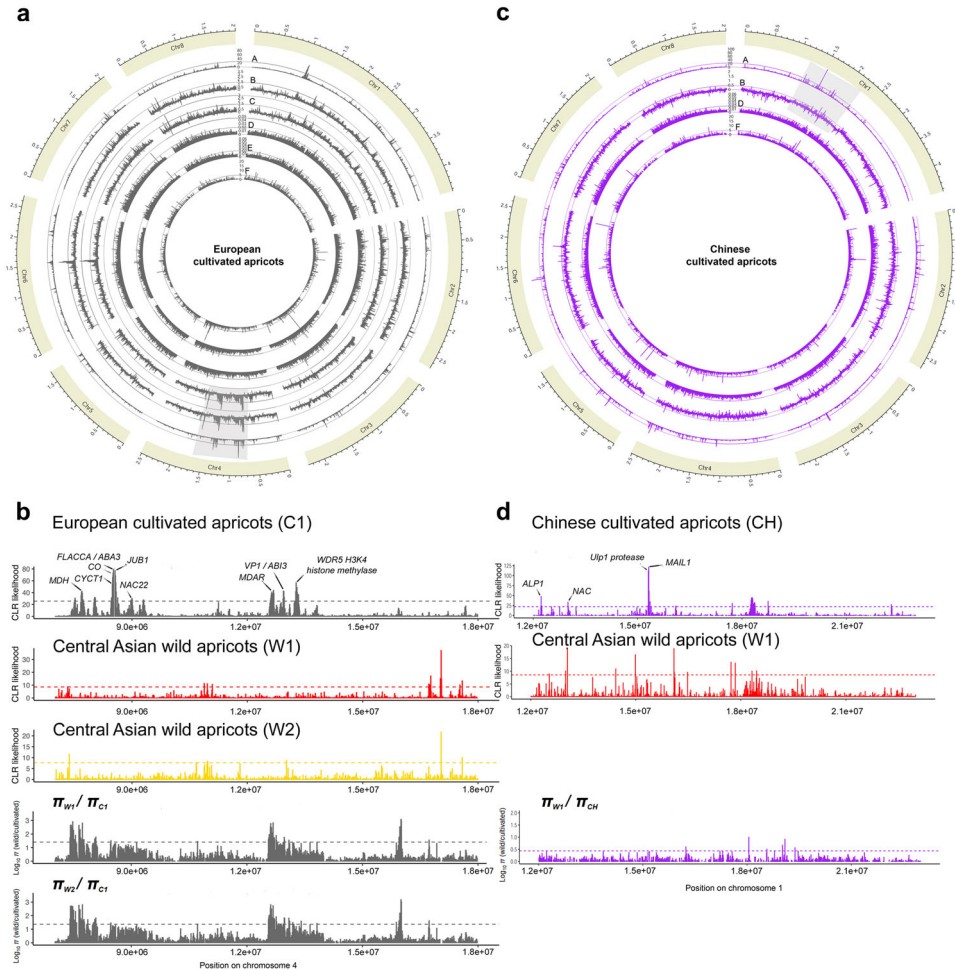

**Fig. 6 Detection of positive selection in European and Chinese apricot genomes.** Density of genes under positive selection along the European (**a**) and Chinese (**c**) cultivated apricot genomes. The different tracks are: A- Composite likelihood ratio (CLR) values, B- $\pi_{W1}/\pi_{Cultivated}$ ratio, C- $\pi_{W2}/\pi_{Cultivated}$ ratio, D- $D_{XY}$ pairwise values between W1 and cultivated apricots, E- $D_{XY}$ pairwise values between W2 and cultivated apricots; F- Linkage disequilibrium LD calculated with Omega. Shaded squares correspond to the intervals depicted in close-up in (**b**) and (**d**). Broken lines indicate the Top 0.5% threshold. Arrows indicate target genes for selection as discussed in the manuscript. W1 in red and W2 in yellow: wild Southern and Northern Central Asian *P. Armeniaca*, C1 in grey and CH in purple: European and Chinese cultivated *P. armeniaca*, respectively. Source data are provided as a Source Data file.

Using the nucleotide diversity ratio ($\pi_{Wild}/\pi_{Cultivated}$), we found higher ($p < 0.05$ by Wilcoxon Signed-Rank test) mean nucleotide diversity in Southern ($\pi = 4.75e\text{-}3\mp2.90e\text{-}3$) and Northern Central Asian wild apricots ($\pi = 4.97e\text{-}3\mp3.01e\text{-}3$) than in the European cultivated apricots ($\pi = 3.29e\text{-}3\mp2.33e\text{-}3$). The European cultivars retained about 66% of nucleotide diversity in comparison with the wild population genetically closest to its progenitor ($\pi_{W1}/\pi_{C1} = 1.44$, $\pi_{W2}/\pi_{C1} = 1.51$). Chinese cultivars displayed higher mean nucleotide diversity ($\pi = 5.34e\text{-}3\mp3.17e\text{-}3$) than their closest related population, the Southern Central Asian wild apricots ($\pi_{W1}/\pi_{CH} = 0.89$), as expected based on admixture footprints.

*Different pathways targeted by selection during European and Chinese domestication events.* Within the selective sweep regions detected with CLR, we predicted 2,018 genes for the European cultivars and 1,252 genes for Chinese apricots, which correspond to 5.3% and 3.3% of the transcribed apricot genome, respectively (Supplementary Data 21). The McDonald Kreitman test run on European and Chinese apricots identified 232 and 44 genes, respectively, as evolving under recurrent positive selection. The set of 2,018 genes within selective sweeps in European apricots were significantly enriched in the glutathione metabolic process,

gene silencing by RNA and triterpenoid biosynthetic process (Supplementary Data 22). Glutathione plays a critical role in maintaining the redox poise under environmental constraints in plants, including trees and fruits[48]. Among the genes within the top 0.5% most significant values of CLR, the functions molybdenum-linked biosynthesis, malate metabolic process and regulation of cyclin-dependent kinase activity were the most enriched. Regarding MK tests, enzymes linked to malate transport appeared to accumulate more non-synonymous mutations than expected under neutrality (Supplementary Data 22). These three biological processes are essential for plant growth and most particularly during fruit development and ripening[49]. A cluster of three molybdenum-related genes (*FLACCA/ABA3*) displayed signatures of selection in European apricots (CLR test), and a clear geographical distribution of selected haplotypes (Figs. 6b, 7). Malate, together with citrate, are crucial for fruit acidity and fruit development[50]. Several NADP-malate dehydrogenase (MDH) encoding genes, mapping on chromosomes 4 (Figs. 6b and 7) and 7, showed signatures of selection either through the CLR, MKT and $\pi$ (Supplementary Data 20 and 24). Three copies of the aluminum-activated malate transporter (ALMT) on chromosomes 2 and 5 were enriched in non-synonymous mutations (Supplementary Data 24). The position of one copy overlapped

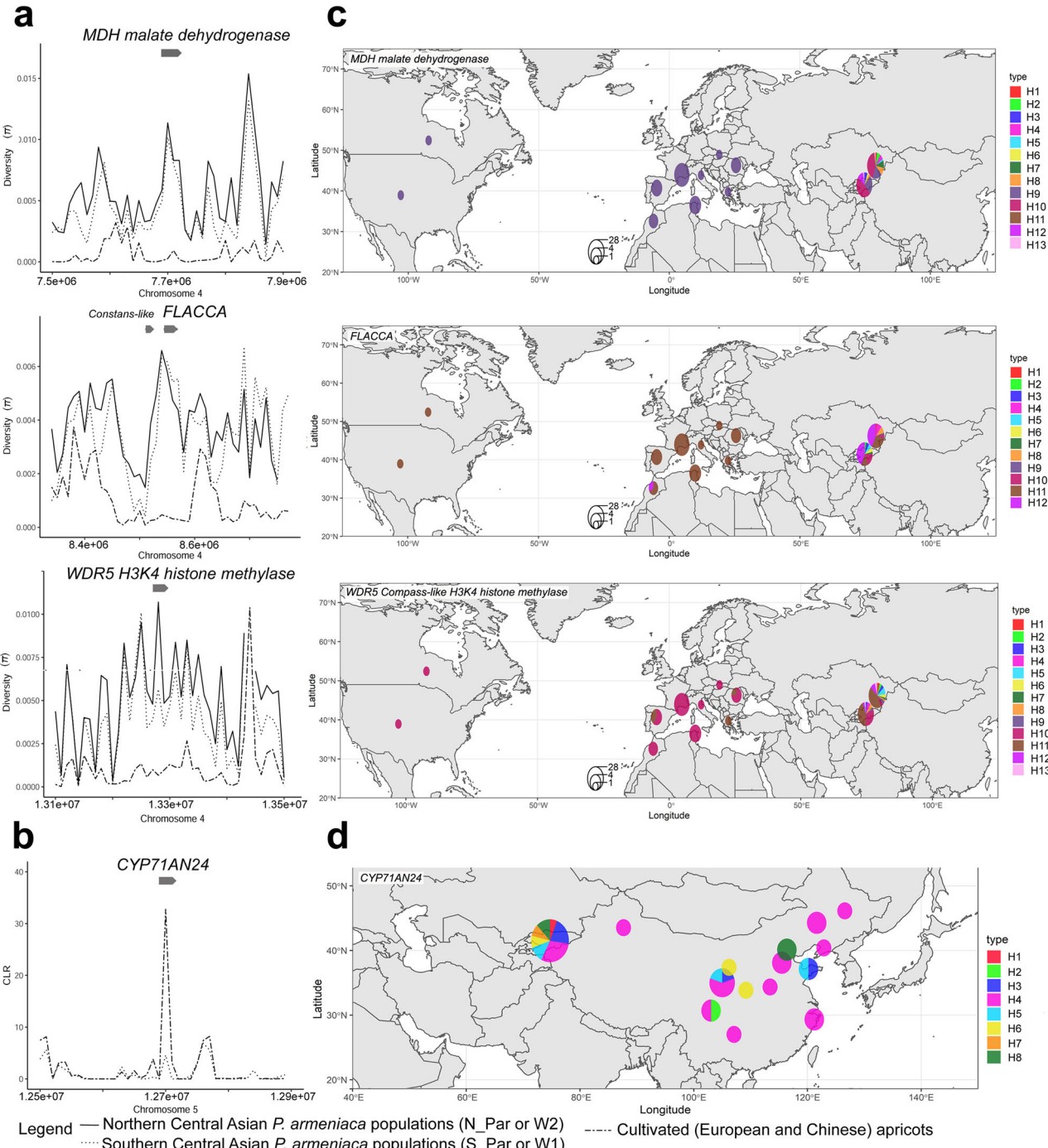

**Fig. 7 Examples of four genes with footprints of positive selection.** Nucleotide diversity (**a**) composite likelihood ratio (**b**) and haplotype distribution of four candidate genes under selection during the domestication of European (**c**) or Chinese (**d**) apricots. **a** Nucleotide diversity (π) plots and (**c**) geographic distribution of alleles at malate dehydrogenase *(MDH)*, *FLACCA* and WD-repeat protein-5 (*WDR5*) loci in European apricots and Northern and Southern Central Asian *P. armeniaca* natural populations. **b** Composite likelihood ratio and (**d**) geographic distribution of alleles at the cytochrome P450 71AN24 (*CYP71AN24*) locus in Chinese apricots and Southern Central Asian *P. armeniaca* natural populations. Grey arrows at the top of the plots indicate the position of the candidate genes; black line, Northern *P. armeniaca* wild natural populations (W2 or N_Par); dotted line, Southern *P. armeniaca* wild natural populations (W1 or S_Par); broken line, cultivated European or Chinese apricots. See Fig. 6b for diversity scans across the whole chromosome 4. Note that the *Constans-like* haplotype distribution is similar to the *FLACCA* haplotype distribution, being very close (22 Kb apart) over the interval. H1 to H13 refer to the haplotypes identified in wild and cultivated apricots. For all panels, the different haplotypes of the focal gene are represented by different, arbitrary colors. The world map layout was generated by the ggplot2 R package. Source data underlying Fig. 7c and 7d is provided as a Source Data file.

with a major locus on chromosome 2 controlling apricot fruit acidity[51]; an ortholog was shown to be under selection during apple domestication based on $\pi$ and $F_{ST}$[18] and a premature stop codon in one of the ALMT genes was associated with lower apple fruit acidity[52]. In European cultivated apricots, homologs of cyclins T1 and KRP1 (cyclin-dependent kinase inhibitor) on chromosome 4 (Fig. 6b) and of CDK on chromosome 6 displayed signatures of positive selection, either by CLR or MKT (Supplementary Data 20 and 24), whose functions are involved in mitotic cell division rate[53–55]. Functional enrichment analysis for selective sweeps identified with Tajima's D, $\pi$ ratio ($\pi_{Wild}/\pi_{Cultivated}$) or LD also highlighted components of cyclin-dependent kinase activity and molybdenum-linked biosynthesis (Supplementary Data 22) and many of the above candidate genes (MDH, cyclin and FLACCA molybdenum cofactor sulfurase) were also identified either by Tajima's D or $\pi$ ratio (Supplementary Data 24). Altogether, these findings indicated that artificial selection during European apricot domestication targeted increased cell expansion and fruit size as well as lower acidity. It also provides valuable clues for scientists to address the nature of interaction between size and composition during apricot selection by humans. Contrary to what was shown in grape and pear[56,57], we did not identify in European apricot genomes signatures of selection for genes directly involved in sugar metabolism, but rather genes that regulate sugar contents in fruits (Supplementary Data 24). Hence, while sugar accumulation and transport are critical events during grape berry ripening[58], the balance between sugars and acidic compounds appears to be a crucial element of European apricot development and maturation. A substantial fraction of our candidate genes were thus also found under selection or controlling important fruit QTL in other Rosoideae fruit species (Supplementary Data 20 and 24), which further supports their importance and also indicates the potential of translational research among these species.

In Chinese cultivated apricots, the functions of the coding sequences within selective sweeps identified by CLR mostly corresponded to repeat and transposable elements (Supplementary Data 23) and may thus correspond to selection in distant regulatory regions. Alternatively, the lack of identified functions other than transposable elements could be due to the genes under positive selection in Chinese apricots being missing in the reference Marouch #14 genome, as previously found in rice when using the domesticated rice IRGSP 4.0 genome as a reference[59]. In addition, we identified many candidate regions harboring resistance or defense-related genes whose functions are not classified as such in the GO analysis. We therefore compared the proportion of genes with NBS, LRR and/or TIR domains among the genes under positive selection (CLR and MKT) and in the whole genome, and found a significant enrichment for such resistance genes in Chinese apricots (Chi squared test, $p$ value=1.78E–22): 11% of resistance genes among those under positive selection (33 out of 301 annotated genes) in Chinese cultivated apricots and 3% in European cultivated apricots (15 out of 491) compared to 0.8% in the Marouch #14 genome (320 out of 37,894 annotated genes).

Our results overall indicate that artificial selection mostly affected distinct loci in the European and Chinese cultivated apricots, despite convergent phenotypic traits, and that genes under positive selection appear to be non-randomly distributed among chromosomes in the two domesticated populations.

*Fruit quality and perennial life cycle traits have been the main targets during apricot domestication.* Based on the annotation of the genes with footprints of positive selection (Supplementary Data 19 and 20), it appears that major fruit traits were most specifically targeted by humans during apricot domestication

before or after diffusion to Europe (and to a lesser extent, during Chinese domestication): fruit acidity, fruit size and yield, firmness, ripening, and fruit flavors (Supplementary Data 24). Many of them were located on chromosome 4 (see above and Supplementary Note 14) but not exclusively. Interestingly, differences in fruit size between European cultivated and wild Central Asian apricots have been previously documented, together with other fruit-related quality traits for Central Asian apricots such as higher yield and sugar contents, lower acidity and increased abiotic stress tolerance[60]. However, cultivated apricots are not only used for fresh consumption but also for fruit drying before consumption. We identified signatures of selection among the top 0.5% scores in both European and Chinese cultivated apricots over genes linked to post-harvest softening, cell wall metabolism and post-harvest pathogen resistance (Supplementary Data 24). While dried apricot has been historically consumed in Central-Asian and Irano-Caucasian civilizations, the apricot kernel was favored in China[61]. In the closely related species *P. dulcis* (almond), the sweet vs. bitter taste of kernels has been linked to lower expression of two genes encoding cytochrome P450 enzymes, CYP79D16 and CYP71AN24 that control the cyanogenic diglucoside amygdalin pathway[62]. We identified significant signatures of selection with the likelihood method (top 0.5% scores) on one of those loci, CYP71AN24, located on chromosome 5 (Fig. 7b-d), but only in the Chinese apricot genomes (Supplementary Data 24).

Beside fruit traits, the temperate perennial fruit tree life cycle differs from that of annual fruiting species in the timing control of the establishment, the onset and finally the release of vegetative rest, i.e., dormancy. This biological process allows alternating active growth, reproduction and vegetative break, following seasonal changes (temperature, day-length) in climate conditions. The fine-tuning of this biological process determines the fitness of temperate perennials. The molecular control of growth cycle includes the control of flowering time, circadian cycles, leaf senescence and adaptation to variable level of winter chilling[63]. The genes identified in regions evolving under positive selection (MKT and CLR-detected) were enriched, both in European and Chinese apricots, in genetic factors controlling circadian clock, growth arrest and leaf senescence including the central longevity regulator, JUNGBRUNNEN 1 (Supplementary Data 20 and 24), suggesting selection on tree phenology, to enhance production or for local adaptation. We also identified overlaps between selective sweeps and known chilling requirement and flowering QTLs[64]: WDR5 COMPASS-like H3K4 histone methylase ortholog on chromosome 4 that epigenetically controls the Flowering Locus C in *Arabidopsis thaliana* (Fig. 6a, Fig. 7)[65] and a serine/threonine protein kinase WNK/with no lysine(K) on chromosome 2 that regulates flowering time by modulating the photoperiod pathway[66] (Supplementary Data 24). Besides those two promising candidate genes, regions with signatures of positive selection were also enriched for key factors of the epigenetic and/or photoperiodic control of flowering, such as a *CONSTANS-like* gene (Fig. 7a), a central regulator of the photoperiodic pathway, triggering the production of the mobile florigen FLOWERING LOCUS T that induces flower differentiation[67] (Supplementary Data 24). A substantial fraction of our candidate genes were thus also found under selection or controlling important fruit QTL in other Rosoideae fruit species (Supplementary Data 20 and 24), which further supports their importance and also indicates the potential of translational research among these species.

## Discussion
Because of its relatively small sized diploid genome and the availability of wild gene pools, apricot can be considered as a

good model to study the genome-scale evolutionary consequences of perennial fruit crop domestication. Based on morphological and botanical data, apricot had long been considered to have originated from China[29]. In the current study, we showed that the European cultivated apricots derived from the Northern Central Asian wild population while the cultivated Chinese apricots were domesticated from the Southern Central Asian wild population. Such independent events of domestication of fruit crops in Europe and Asia have also been reported in pears[56]. The dates of domestication events were estimated to be ca. 2,000-3,000 years ago, which is consistent with archeological data. In Central Asia, apricot cultivation began around I–II millennia BC[68,69] and modern excavations in southern Turkmenistan and Uzbekistan indeed did not find evidence for the use of fruit and nuts in western Central Asia before 1500 BC (Before Christ)[70]. In contrast, apricot kernels have been found in China in relics of the Zhumadian city (Henan province), dating from the Xia period (2070–1600 BC)[71]. Other apricot archeological remains were also found in Jingmen city (Hubei province), during the excavation of the tomb of Chu in Baoshan, dating from the Warring States period (475–221 BC)[72]. We also showed in the current study that Chinese cultivated apricots had higher nucleotide diversity than its wild Central Asian closest relatives (112%) while European apricots had lower diversity, although still relatively high (66% compared to its wild Central Asian closest relatives). This suggests a more severe loss of diversity, i.e. a stronger bottleneck, during European apricot domestication than Chinese apricot domestication, and also concurs with the view that domestication bottlenecks are less severe in perennials than in annuals due to higher rates of outcrossing and higher population effective sizes[14]. Perennial fruit crops maintain an average of ~95% of the neutral variation found in wild populations, as shown in apples[18,73]. Only peach appeared as an exception, for which only 34% was retained in Landraces and 25% in Western cultivars[74]. Annuals in contrast retain on average ~60% of their progenitor variation[14]. The higher genetic diversity in Chinese cultivated apricots can be explained by: (i) the lower fraction of self-compatible accessions in Chinese cultivated apricots (10%[37] than in European apricot cultivars 51–58%[38,39]); (ii) a higher degree of gene flow with wild relatives in Chinese than in European cultivated apricots. European apricots originated from Central Asia and were later disseminated westwards to Europe where no recent wild-to-crop admixture occurred, except sporadically with wild plum (giving rise to the black or purple apricot[75]). A recent study on the single wild European Armeniaca species, P. brigantina, found no signature of admixture between the cultivated apricot germplasm and its cross-compatible wild relative[27]. In China in contrast, at least three Armeniaca wild related species share habitats and hybridize with cultivated apricots, i.e., P. sibirica in the North, P. mandshurica in the NorthEast and P. mume in the South. Past hybridization and ongoing gene flow between P. sibirica and P. armeniaca were illustrated in the current study, but only in the Chinese germplasm. As examples of documented wild-to-crop introgression in China among Armeniaca species, we can also cite the sweet kernel apricot (a hybrid between P. sibirica and P. armeniaca which is used for traditional Chinese medicine purposes[61]), P. mume[76] and the Apricot Mei (a hybrid between P. mume and P. armeniaca)[31]. More generally, hybridization has often played a central role in the origin and diversification of perennials, leading to adaptation to new environments after dispersal[13,77]. In apple in particular, the cultivated Malus domestica germplasm results from an initial domestication from the Asian wild apple M. sieversii followed by introgression from the European crabapple M. sylvestris[73].

In addition to elucidating the evolutionary history of Armeniaca wild species and of the cultivated apricots, with two

independent domestication events from different wild populations, we also identified footprints of positive selection. As expected for perennials[13], we found that a small part of the genome has been affected by selection (0.42% and 0.22% in European and Chinese apricots, respectively). Selection footprints appeared more abundant in European apricots, with a hotspot on chromosome 4, while admixture was much more pervasive in Chinese cultivated apricots. This difference in the fraction of genomic regions showing signatures of selection between European and Chinese cultivated apricots reflects either a more limited effect of human selection during the domestication of Chinese apricots or a counter-effect of gene flow on the reduction of genetic diversity by selection in Chinese apricots. In both cultivated groups, the genes affected by selection had predicted functions associated with perennial life cycle traits, fruit quality traits and disease resistance, as expected for traits likely under selection during fruit tree domestication. Some of these candidate genes colocalized with previously identified genomic regions[46,47,51,78–80]. Essential target traits of domestication in fruit crops likely include fruit size, sweetness, ripening and texture, tree architecture as well as flower and fruit phenology. Another key trait likely associated with adaptation of cultivated apricot trees is winter chill requirement that determines flowering time[81]. These functions under selection appear strikingly similar to those in domesticated apple, peach and pear trees in which selective sweeps pointed to genes also associated with fruit sugar content, size, firmness, color, shape, flavor and/or acidity[56,82,83]. The traits under selection in fruit crops were thus as expected different from those in annual crops, in which the traits under selection are often the loss of seed shattering, the minimization of seed dormancy and an increase in seed size and number[8]. We showed that, despite phenotypic convergence between European and Chinese cultivated apricots, different genomic regions and different functions were affected by selection, as also found in pears[56]. This indicates that different genomic changes can lead to the same adaptive phenotype, concurring with previous studies on annual crops[8,9], as well as natural populations[84,85]. In addition to fundamental knowledge on the processes of adaptation, our study identifies genomic regions of high importance for fruit tree breeding.

## Methods

**Plant material.** Whole-genome sequences from a total of 926 individual trees were analysed: 184 cultivated apricots (P. armeniaca) with different geographical origins, 258 wild P. armeniaca from 14 Central Asian natural populations, 43 P. sibirica, four P. mume, one P. mandshurica and fourteen P. brigantina, one peach (cv. Honey Blaze) and one almond (cv. Del Cid) outgroups. We also included 348 P. mume genomes and 72 apricot cultivars reported in previous work[31,33]. Two apricot cultivars were selected for obtaining high-quality genome assemblies, the Marouch #14 accession for its high level of homozygosity and Stella cv. as a main source of resistance to sharka disease[33]. Two Chinese accessions were also selected for genome assembly as representatives of the P. sibirica (CH320.5) and P. mandshurica (CH264.4) species, respectively. Details on the 578 sequenced Prunus genomes are available in Supplementary Data 1 and Supplementary Note 1.

**Illumina sequencing, PacBio and nanopore library construction, sequencing and optical genome maps construction.** Methods for DNA/RNA preparation, short- and long-range sequencing and optical map constructions are available in Supplementary Note 2.

Marouch #14 and cv. Stella genome assemblies, error correction and phasing were performed with FALCON/FALCON-Unzip v0.7 from PacBio long-reads[32] (Supplementary Fig. 1). A hybrid assembly was then produced by using a Bionano Genomics optical map (Supplementary Note 3). To further improve these assemblies, we used ILLUMINA short reads to perform gap closing. Ordering and orientation of genomic scaffolds to reconstruct chromosomes were performed using molecular markers as described in Supplementary Note 4. A complete list of all primers used, including the names and sequences, is available in Supplementary Data 6.

Several genome assemblies were generated for CH320_5 and CH264_4 (Supplementary Note 3). We selected for each of the two accessions the assembly

obtained using SMARTdenovo with all raw reads[86]. Assemblies were polished using both long and short reads (with Racon and Pilon respectively)[87,88], and contigs were organized using optical maps (Supplementary Note 3). Negative gaps were closed using BiSCoT[89] and the consensus was polished using Hapo-G[90], a polisher dedicated to heterozygous genome assemblies. The quality of the genome assemblies was assessed as described in Supplementary Note 4.

**Annotation of protein-coding genes and transposable elements**. Protein coding genes were annotated using a pipeline integrating the following sources of information: i) a BLASTp search of reciprocal best hits; (ii) EC (Enzyme Commission) numbers; (iii) the transcription factors and kinases; (iv) the Interpro (release 81.0) and BLASTp hits against NCBI NR database restricted to Viridiplantae proteins as input datasets for Blast2GO annotation service to produce functional descriptions and gene ontology terms. Repetitive elements were predicted in the four Armeniaca genomes assembled in this study using REPET package v2.5 (https://urgi.versailles.inra.fr/Tools/REPET)[91] (Supplementary Note 5).

**Comparison with previously assembled *P. armeniaca* genomes**. We downloaded the three existing assemblies from the Rosaceae genome database (cv. Chuanzhihong[35]) and from NCBI (cv. Rojo Pasion[36]). Contigs were obtained by splitting the scaffolds at each gap (of at least one N), and gene completion was calculated using BUSCO (v4.0.2 with default parameters)[92] and the eudicotyledon odb10 database ($N = 2,121$ genes). Whole genome alignments were performed using minimap2 (version 2.15 with default parameters[93]) and dotplots were generated from alignments larger than 5Kb using dotPlotly (https://github.com/tpoorten/dotPlotly).

**Whole genome alignment and variant calling**. The assembled genomes of cv. Stella, CH320_5 and CH264_4 were aligned to the reference Marouch #14 reported in this work using the runCharacterize script provided by Bionano Genomics, with the default settings. The genome alignments were imported into Bionano Access software for visualization (Supplementary Note 6).

The assembly alignments obtained above were used to call structural variants using the runSV script provided by Bionano Genomics, with default settings. The smap file resulting from this analysis was filtered out to extract the insertions, deletions, inversions, duplications and translocations. The structural variations can be visualized into Bionano Access software. The R package OmicCircos was used to edit the circos plot figure from the filtered smap file.

**Phylogeny and reconstruction of ancestral chromosomal arrangements of Armeniaca species**. We identified only 298 single-copy orthologous genes shared among the 12 following species: *Arabidopsis thaliana*, *Populus trichocarpa*, *Vitis vinifera*, *Rosa chinensis*, *Fragaria vesca*, *Prunus persica* cv. Lovell, *P. dulcis* cv. Texas, *P. mume*, *P. mandshurica*, *P. sibirica*, *P. armeniaca* Marouch #14 and *P. armeniaca* cv. Stella (Supplementary Data 10). Fourfold degenerate sites (4DTv) from these 298 single-copy orthologous genes were extracted and concatenated into a "supergene" format for each species. The 12 aligned fourfold degenerate site supergenes were used to construct a phylogenetic tree using the BEAST software[94] (Supplementary Note 7). The Armeniaca chloroplast phylogeny was inferred as detailed in Supplementary Note 8 and the evolutionary scenario of genome chromosomal arrangement was inferred based on synteny relationships identified between the Armeniaca genomes and other Rosaceae genomes[44] (Supplementary Note 7; Supplementary Data 10).

**Sequence alignment and variation calling**. ILLUMINA sequence reads for each accession were mapped to the Marouch #14 genome (Supplementary Note 9). Reads were filtered for low mapping quality (MQ < 20) and by removal of PCR duplicates (Supplementary Data 1). Both paired-end and single-end mapped reads were used for SNP detection throughout the entire Armeniaca accessions in the GATK toolkit (version 3.8)[95] (Supplementary Note 9). A subset of 15,111,266 SNPs was selected after filtering for bi-allelic SNPs, SNP quality (>30) and missing data (< 15 %).

**Linkage disequilibrium analysis**. We quantified LD using the squared correlation coefficient ($r^2$) between pairs of SNPs along 300 Kb windows as implemented in PLINK v1.9[96]. An average of 50,000 SNPs were randomly selected from each chromosome. The decayed physical distance between SNPs was identified as the distance at which the maximum $r^2$ dropped by half (averaged in short range of 10 bp)[97] (www.cog-genomics.org/plink/1.9/) (Supplementary Note 9). Raw SNP data was further filtered by vcftools [—max-missing 0.85—maf 0.05/0.01—minQ 30], and LD pruned in PLINK v1.90[96] [—indep-pairwise 50 5 0.0428].

**Population subdivision**. We investigated the occurrence of gene flow among populations using the ABBA-BABA test implemented in *D*-suite[98,99] (Supplementary Note 10) and the parentage relationship between Armeniaca accessions by identity by descent (IBD) in PLINK v1.90[96] (Supplementary Note 11). The fastSTRUCTURE software (version 1.0) was used to infer the Armeniaca population

structure[100]. We ran fastSTRUCTURE on four datasets: (1) the whole Armeniaca dataset made of 917 individuals (after removal of the other Prunus species, outside of the Armeniaca section, and of interspecific hybrids), (2) *P. mume* ($N = 348$), (3) the rest of the Armeniaca species, without *P. mume*, that were sequenced in the current study (later referred as the set of individuals of other Armeniaca species, $N = 555$) and (4) the set of 202 unique and non-admixed accessions (Supplementary Notes 11 and 12). fastSTRUCTURE was run on a subset of 95,686 MAF > 0.05 filtered and LD pruned SNPs for the first three datasets while 9,613 SnpEff-filtered[101], synonymous SNPs were used for the last dataset of 202 unique, non-admixed accessions. This dataset was also used for demographic inferences and the average number of nucleotide differences between genotypes ($\pi$) was drawn from each population[102] using pixy[103] and Stacks[104]. Other summary statistics ($H_E$, $H_O$, $F_{IS}$, number of private alleles) were computed with Stacks[104]. Principal component analysis was performed using the smartPCA program of the EIGENSOFT package (version 6.1.4) in R software environment[105].

**Inferences of demographic and divergence histories**. We used random forest approximate Bayesian computation[106] to unravel the evolutionary history of the cultivated and wild apricots. From the inferred population structure for $K = 7$ including the 202 Armeniaca unique accessions (Supplementary Note 12), we filtered out admixed individuals (*i.e.*, individuals with a membership coefficient < 0.90 to a given genetic cluster). A total of 163 non-admixed unique accessions were therefore used for ABC-RF inferences, which included six genetic groups: 25 European (C1) and 10 Chinese (CH) cultivated accessions, 33 and 43 Central-Asian accessions from W1 and W2 *P. armeniaca* natural populations, respectively, 23 wild *P. sibirica* from the W4 genetic cluster and 29 *P. mume* individuals (Fig. 5a, Supplementary Note 12). Four ABC-RF steps were then used to infer the most likely scenarios of domestication of cultivated and wild apricots (Supplementary Note 12; Supplementary Fig. 28).

**Selective sweep identification**. We looked for patterns of selective sweeps in the European (C1, $N = 50$) and Chinese (CH, $N = 10$) populations and their wild progenitors (W1 and W2, $N = 33$ and 43, respectively). Multiple types of SFS (site frequency spectra) derived, LD patterns and neutral index tests were used to detect positive selection, and differentiation between cultivated populations and their genetically closest wild population(s) (Supplementary Note 13). Composite-likelihood ratio tests (CLR) were run with the SweeD software (version 3.0)[107] and LD-$\omega$ tests were performed with Omegaplus (version 2.0)[108]. The McDonald-Kreitman test (MKT), $F_{ST}$, $D_{XY}$, and Pi ($\pi$) tests were computed with the Popgenome R package (version 2.7.5)[109]. The Tajima's D was computed using vcftools (version 0.1.16)[110]. The significance of the selective sweep signals were inferred using different scaling thresholds (Supplementary Note 13).

**GO enrichment and candidate gene analyses**. Gene ontology (GO) enrichment analyses were performed for the candidate genes present within selective sweep intervals using GO annotation terms (biological process, molecular function and cellular component) extracted from the annotated Marouch #14 genome by using Gprofiler2 and filtered with the Benjamini-Hochberg multiple test correction ($p$-adjusted <0.05). Genes present within the most significant selective sweeps were retrieved from Marouch #14 gff3 file using the coding sequence coordinates (Supplementary Note 14). Principal component analysis was performed using the 'smartpca' programme from EIGENSOFT software version 7.2.1[105], after merging the vcf files for each candidate gene independently. Based on the first ten PCs, we grouped individuals using hierarchical clustering (Euclidean distance and Ward method) and drew optimal partitions using the factoextra R package (https://www.R-project.org/). Haplotype distribution was visualized geographically using the 'scatterplot3d' R package[111] (Supplementary Note 14).

**Reporting summary**. Further information on research design is available in the Nature Research Reporting Summary linked to this article.

## Data availability

Data supporting the findings of this work are available within the paper and its Supplementary Information file. A reporting summary for this Article is available as a Supplementary Information file. All the raw sequencing data generated during the current study were deposited in the European nucleotide archive (ENA) under the following accession numbers: PacBio de novo sequencing of *Prunus armeniaca* Marouch #14 and *Prunus armenica* cv. Stella-PRJEB42606; Oxford Nanopore Technologies de novo sequencing of CH320_5 and *P. mandshurica* CH264_4-PRJEB40668; Illumina DNASeq paired-end reads—PRJEB42181 and PRJEB40984; Illumina RNASeq paired-end reads-PRJEB42479. De novo genome assemblies and annotation are deposited to the Genome Database for Rosaceae and are available under the links: *Prunus armeniaca* Marouch n14 whole genome v1.0 assembly & annotation [https://www.rosaceae.org/Analysis/9642068]; *Prunus armeniaca* cv. Stella whole genome v1.0 assembly & annotation [https://www.rosaceae.org/Analysis/11326140]; *Prunus mandshurica* CH264_4 whole genome v1.0 assembly & annotation [https://www.rosaceae.org/Analysis/10024324]; *Prunus sibirica* CH320_5 whole genome v1.0 assembly & annotation [https://www.rosaceae.org/Analysis/9955981]. Source data are provided with this paper.

## Code availability

Scripts used for analyses are available at GitHub [https://github.com/CornilleAmandine/-apricot_evolutionary_history_2021]. Tools for genome assembly and population genomic analyses are given in the Supplementary Note 15.

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

## Acknowledgements

Most of the computational resources and infrastructure used in present publication were provided by the Bordeaux Bioinformatics Center (CBiB). Additional computer time for this study was provided by MCIA (Mesocentre de Calcul Intensif Aquitain) of the Universities of Bordeaux and of Pau and des Pays de l'Adour. We are also grateful to the Genotoul bioinformatics platform, Toulouse, for providing help, computing and/or storage resources (http://bioinfo.genotoul.fr/index.php) and the URGI platform (https://urgi.versailles.inra.fr/Tools/REPET), for help in running the REPET package v2.5. We acknowledge valuable contribution of Dr Peter Civan (INRAE GDEC, Clermont-Ferrand) for chloroplastic phylogenomics and admixture analysis and Dr Ricardo Rodriguez de la Vega (Université Paris-Saclay, ESE, Orsay) for help in 'managing' the BEAST. We thank the INRAE BFP technical team: Aurélie Chague for the extraction of *Prunus* genomic DNA for ILLUMINA sequencing, Mélodie Caballero for the extraction of RNA, Jean-Philippe Eyquard and Pascal Briard for the care of the plants. S.L. was supported by the Chinese Scholarship Council (2015-2019), X.C. by the ATIP CNRS Inserm funding. This research was

supported by the ABRIWG project (ANR CHEX 2012-2014), by Genoscope, the Commissariat à l'Energie Atomique et aux Énergies Alternatives (CEA), France Génomique (ANR-10-INBS-09–08, SWAG project), Bordeaux University (G2P SWAGMAN and ATT ABXING), INRAE Biology and Plant Breeding division (WildArm project). This work was performed in collaboration with the GeT core facility, Toulouse, France (http://get.genotoul.fr), and was supported by France Génomique National infrastructure, funded as part of 'Investissement d'avenir' program managed by Agence Nationale pour la Recherche (contract ANR-10-INBS-09) and by the GET-PACBIO program (« Programme opérationnel FEDER-FSE MIDI-PYRENEES ET GARONNE 2014-2020 »). The ancestral karyotype reconstruction approach was supported by the Institut Carnot Plant2Pro (#0001455 project SyntenyViewer 2017) and the ISITE CAP2025 (#00002146 SRESRI 2015 'Pack Ambition Recherche Project' TransBlé 2018). The Bergeron x Bakour genetic maps were constructed in the frame of the ABRIWG CHEX ANR (2012-2014) and of the Resibac CASDAR (2013-2016) projects. The Liaoning pomology institute benefited from Grant/Award Number: 2019YFD1000600 from the National Key Research and Development Program of China.

## Author contributions

V.D., A.G., and A.G.A. initiated the study and V.D. coordinated the efforts. V.D., T.G., A. C., S.L., A.G.A., and A.G. conceived and designed experiments. A.C., S.L., X.C., and S.D. carried out all population genetic analyses. S.L. and S.D. were responsible for running the selection and differentiation tests; A.G. performed GO enrichment; V.D. and P.P. analysed the raw selective sweep data. A.G. and J.S. performed the phylogenetic studies. A.G., A.B., M.N. performed all genome assemblies, together with J-M.A., B.I., C.B. and C.C. S. A., W.M., N.R., W.R., and S.C. were responsible for the optical maps and identified structural variants. R.D. and E.D. prepared high quality apricot DNAs for long-range sequencing. O.B., C.L-R., and R-F.S. performed PacBio and RNAseq sequencing. Q.T.B. performed TE annotation and A.G. and S.C., gene annotation. Q.T.B., J.T., and S.L. performed candidate gene analyses. J-M.A., G.R., T.Z., W-S.L., and P.L. contributed with plant material, genetic and morphological data. V.D. and D.T. built the botanical collection; D.T. is running it. V.D., T.G., A.C., and A.G.A. wrote the paper. All authors commented on the manuscript.

## Competing interests

The authors declare no competing interests.
