## [Peer Review File · Nature Communications]

REVIEWER COMMENTS

Reviewer #1 (Remarks to the Author):

This paper describes high- quality genome assemblies of cultivated apricot and immediate wild relatives, population genomic analysis of a large collection of individuals and also selection analyses of domesticated apricots.

This is an excellent large-scale analysis of this important fruit tree species. The assemblies will be of great interest to the plant genome community and the analyses provide key information on the evolution of this fruit tree crop. The analyses are well-considered and well-executed and the results will be of wide interests.

I only have two suggestions:

1. The dating of the domestication of apricots using genome analysis is interesting. It would help improve the Discussion if the dates are placed in the context of archaeological findings regarding apricot domestication – I am sure there are examples in the literature. That will help place the dates in a good context.
2. I think the in-depth discussion of various genes and processes in the selection analyses is premature – I am always worried about such discussions in the absence of more in-depth functional information. These types of discussions end up being very speculative. I would suggest deleting much of this results on possible genes and traits and maybe just mentioning (very briefly) one or two genes of interest. That will reduce the amount of speculation on possible domestication genes and focus simply on reporting the results of the selective sweep analyses.

Reviewer #2 (Remarks to the Author):

The authors of this study present a domestication genomics study of apricot by analyzing the genomes of ~600 trees including four high-quality assemblies of divergent apricots. The population structure revealed that Chinese and European apricots result from two different gene pools from independent domestication events. Footprints of selection in these two groups were analyzed and found to be different between the two sets, however, shared related functions of genes which appeared under selection. This paper presents data and analysis on a scale that has not been presented before in *Prunus* species. The paper is well written. I fully support publication of this manuscript in *Nat Comm*.

There are some points that I labelled "major", though none of these points is raising doubts on the conclusions of the paper. Mostly these are questions concerning methods and their accuracy, which hopefully help to make the manuscript more consistent.

Major:

- 1) The SV calling is based on the artificial translation of sequence to pseudo-optical maps. How does this affect SV calling accuracy and precision of the SVs? Why is this not done on sequence level?
- 2) The four assemblies show considerable differences to published assemblies (Figure S2), but are quite similar to each other (Figure S9). Though this could be due to highly divergent cultivars that were used to generate the cultivar presented by Campoy, but it could also be due to the fact that the haplotypes were arranged after a reference sequence? What is the effect of the sparse genetic map? Is this sufficient for scaffolding?
- 3) Figure 2a. Error margins are missing. Could imprecise estimation (as typically shown with error margins) explain the extreme divergence dates between the cultivars that are suggested by this figure

(which are clearly too large)?

4) Domestication analysis. Removal of samples could affect the site frequency spectrum. The admixed samples could also result from incomplete lineage sorting. Would this lead to different conclusions? Is it possible to estimate the gene flow indicated in Figure 5c?

5) Would the usage of a Chinese apricot as reference lead to other insights on genes that are under selection in the Chinese cultivar?

Minor:

1) Minor but important. The font sizes in the figures are consistently too small, and take away the joy of looking at them.

2) Figure 1d). The amount of structural variation appears too little in number.

3) Line 269-277: numbers do not add up. This could be clearer.

4) Line 346: W1 and W2 are used before being defined in line 366.

5) Figure 5c: Names at the lower part of the figure are not defined.

Reviewer #3 (Remarks to the Author):

Manuscript of "Following the Adaptive Path of Fruit Tree Domestication using Population Genomics: the Case of Apricots" (NCOMMS-21-07239-T) de novo assembled four *Armeniaca* genomes, and re-sequencing 600 apricot accessions to achieve the origin and domestication of apricots. This manuscript provide good materials and large datasets, however, it is not well organized. The main story throughout the MS is not clear. The authors should carefully think about it and reorganize the story. There are some suggestions for this MS.

Line 131. The authors used two sequencing technologies, PacBio and ONT, to sequence the *Armeniaca* genomes. They should compare their advantages and disadvantages on the genome assembly of perennial fruit trees and this will provide useful information for later researchers.

Line 132,138 What is the appropriate sequencing method for different heterozygosity genome? Could you explain your strategy in detail? Why is it 73X and 60X (PB) for Marouch #14 and Stella, and 113X and 139X (ONT for the others)? Have you tried other fold coverage?

Can you gave some introduction of the newly assembled accessions of 'Marouch #14', 'Stella', 'CH320-5', and 'CH264.4'?

Figure 1d Why do the authors use Marouch #14 as the reference genome?

Table 1 'Number of scaffolds *' The unplaced scaffolds could tried to be splited, and then blast to the other references to find the anchored place. If possible, Hi-C data would be an effective way to assist genome assembly.

Line 202. An inversion of ca. 600 Kb was detected in the *P. armeniaca* Marouch #14 genome. This structural variation of 600 Kb should be verified by PCR.

Line 250 Re-sequencing depth of all the accessions varies a lot in supp table S1. Is there any SNP calling bias? How could you deal with the data to avoid it? What is more, there are five candidate genomes. The mapping rate also has a big difference. Is it possible to use the pan-genome as the reference to avoid mapping bias?

Line 249 Have you tried the mitochondrial genome based phylogenetic relationship? How about the

nuclear DNA based phylogenetic relationship? Do they have coincidence?

Line 326 Best K values can be calculated with deltaK.

Line 341 This specification and domestication history would be very interesting, and the authors should be very carefully to verify these results with other strategy, such as genetic diversity methods π , F_{st} , $tajima'D$ and IBD etc..

Line 375. More evidence should be provided for the inference of independent domestication events, for example, identity by descent (IBD) analysis. The author seems to mention this analysis in the method section, however, we have not seen the relevant results to support this inference.

Have you found any differentiation or group-specific mutations for the variation studies?

Southeast part of China will be an important place for plant origin. Have you consider about the sample collection of this region? Comparisons could be made with other published Rosaceae species's origin and evolution history.

The author has used "enriched" or "significantly enriched" many times, please indicate the significant values. In addition, P_value should be P value (Line 519).

Line 527. The authors identified many genes involved in fruit quality and perennial life cycle traits, but they do not explain their expression levels in the main text. The appropriate expression level verification and related discussion should be contained.

In discussion section. The related Rosaceae species domestication process should be discussed.

Line 849. We have not seen the genome data shared by the authors on GDR. The data and annotation information such as gene structure, gene function and TE should be included in it.

Dear editor,

Thank you very much for taking the time to consider our manuscript for publication in Nature Communications and for the constructive review process. We have revised the manuscript taking all reviewers' comments very seriously. We have underlined in yellow the changes in the manuscript. We outline below how we addressed all comments. We hope that the revised version will be satisfactory for publication in Nature Communications and we thank you and the referees for your time and comments.

On the behalf of all authors,

Veronique Decroocq

NCOMMS-21-07239A.

REVIEWER COMMENTS

Reviewer #1 (Remarks to the Author):

This paper describes high- quality genome assemblies of cultivated apricot and immediate wild relatives, population genomic analysis of a large collection of individuals and also selection analyses of domesticated apricots.

This is an excellent large-scale analysis of this important fruit tree species. The assemblies will be of great interest to the plant genome community and the analyses provide key information on the evolution of this fruit tree crop. The analyses are well-considered and well-executed and the results will be of wide interests.

I only have two suggestions:

1. The dating of the domestication of apricots using genome analysis is interesting. It would help improve the Discussion if the dates are placed in the context of archaeological findings regarding apricot domestication – I am sure there are examples in the literature. That will help place the dates in a good context.

>> We thank the referee for the suggestion. We have added a couple of sentences on this matter in the result and discussion parts. In Central Asia, apricot cultivation was introduced around I-II millennia BC (Sinskaya, 1969; Spengler et al., 2018). In accordance with this dating, modern excavations in southern Turkmenistan and Uzbekistan did not find any evidence for the use of apricot fruits or nuts in western Central Asia before 1500 BC (Miller 1999). Botanical evidence for farming in the steppes and mountains of Central Eurasia is only documented after ca 800 BC (Chang et al. 2003; Spengler et al. 2016). This does not preclude that humans sampled, before domestication, fruits directly from the forests, without cultivation. In contrast, apricot kernels have been found in China in relics of the Zhumadian city (Henan province), dating from the Xia period (2,070~1,600 BC) (Zhumadian City Cultural Relics Protection Management Office, 1998). Apricot archeological remains were also found in the Jingmen city (Hubei province), during the excavation of the tomb of Chu in

Baoshan, dating from the Warring States period (475~221 BC) (Baoshan Cemetery Organizing Team of Jingsha Railway Archaeological Team, 1988). Stones of mountain apricot (*Armeniaca vulgaris*) were found at Kangjia (Liu, 2005), dating from Longshan (c. 3,000 – c. 1,900 BC) and Late Shang Dynasty (c.1,400 - c. 1,150 BC). However, this indicates that apricot trees were present in the area, but maybe not domesticated yet.

References :

- Chang C, Benecke N, Grigoriev FP, Rosen AM, Tourtellotte PA. 2003. Iron Age society and chronology in south-east Kazakhstan. *Antiquity* 77, 298–312.
- Sinskaya E.N. (1969) Historical geography of cultivated floras (at the dawn of agriculture). Kolos, Leningrad, USSR (in Russian)
- Miller NF (1999) Agricultural development in western Central Asia in the Chalcolithic and Bronze Ages. *Vegetation History and Archaeobotany* 8: 13–19
- Spengler RN, Maksudov F, Bullion E, Merkle A, Hermes T, Frachetti M (2018) Arboreal crops on the medieval Silk Road: Archaeobotanical studies at Tashbulak. *PLoS ONE* 13(8): e0201409. <https://doi.org/10.1371/journal.pone.0201409>
- Spengler RN, Ryabogina N, Tarasov PE, Wagner M. (2016) The spread of agriculture into northern Central Asia: Timing, pathways, and environmental feedbacks. *The Holocene* 26:1527-1540. doi:10.1177/0959683616641739
- Zhumadian City Cultural Relics Protection Management Office (1998). Zhumadian Yangzhuang—Cultural Relics and Environmental Information of the Upper Huaihe River in the Holocene of China introduction. *Archeology*, (8), 59-59 [Chinese]
- Baoshan Cemetery Organizing Team of Jingsha Railway Archaeological Team, Hubei Province. Brief Report on Excavation of Baoshan Chu Tomb in Jingmen City. *Cultural relics*,1988(5):9 [Chinese]
- Liu, L. (2005). *The Chinese Neolithic: trajectories to early states*. Cambridge University Press.

2. I think the in-depth discussion of various genes and processes in the selection analyses is premature – I am always worried about such discussions in the absence of more in-depth functional information. These types of discussions end up being very speculative. I would suggest deleting much of this results on possible genes and traits and maybe just mentioning (very briefly) one or two genes of interest. That will reduce the amount of speculation on possible domestication genes and focus simply on reporting the results of the selective sweep analyses.

>>We have tried to reduce this part, but we think it is important to briefly explain to readers what importance the genes found within the selective sweeps have. We have tried to make it clearer these are still candidate genes. Moreover, we showed in Extended data S20 and S24 that a substantial number of our candidate genes were also found under selection in other Rosoideae fruit species. This further supports their importance and also indicates the potential of translational research among all those species. We have added a sentence in the ms to clarify this point.

Reviewer #2 (Remarks to the Author):

The authors of this study present a domestication genomics study of apricot by analyzing the genomes of ~600 trees including four high-quality assemblies of divergent apricots. The population structure revealed that Chinese and European apricots result from two different gene pools from independent domestication events. Footprints of selection in these two groups were analyzed and found to be different between the two sets, however, shared related functions of genes which

appeared under selection. This paper presents data and analysis on a scale that has not been presented before in Prunus species. The paper is well written. I fully support publication of this manuscript in Nat Comm.

There are some points that I labelled "major", though none of these points is raising doubts on the conclusions of the paper. Mostly these are questions concerning methods and their accuracy, which hopefully help to make the manuscript more consistent.

Major:

1) *The SV calling is based on the artificial translation of sequence to pseudo-optical maps. How does this affect SV calling accuracy and precision of the SVs? Why is this not done on sequence level?*

>>At that time, the Bionano Genomics tools seemed to be the best tool to detect large structural variations, from 0.5kb to some Mb, at the genome scale. For chromosome 4, we have manually checked the concordance of this structural variation calling with the alignment of the sequence with minimap and with the dot plot with D-genies. Some of the structural variations detected have been validated with the optical maps but could not be visualized with minimap. We have clarified this in the Supplementary note 6. Upon the request of reviewer #3, we have also checked by PCR the inversion event on chr4 and have added this information (see Supplementary Figure S10).

2) *The four assemblies show considerable differences to published assemblies (Figure S2), but are quite similar to each other (Figure S9). Though this could be due to highly divergent cultivars that were used to generate the cultivar presented by Campoy, but it could also be due to the fact that the haplotypes were arranged after a reference sequence? What is the effect of the sparse genetic map? Is this sufficient for scaffolding?*

>>The number of the genetic markers was sufficient to build the pseudo-chromosomes with a high colinearity between the physical position on the chromosome and the genetic maps location as attested by the Pearson correlation coefficient (values closer to -1 and 1 indicate near-perfect colinearity), see Supplementary Figure S5 page 50 of the Supplementary information. We are therefore confident that the differences are genuine. Figure S9 also shows differences between the four genomes we report here, such as gaps and inversions, especially at the ends of the chromosomes. Moreover, the genomes of CH320_5 and *P. mandshurica* (CH264_4) have been finalized by using *P. armeniaca* Marouch #14 as reference as stated in the Supplemental information at the bottom of the page 12 : "Scaffold ordering using comparative genomics: Scaffolds of CH264_4 and CH320_5 were organized into chromosomes using the Marouch #14 v3.1 reference genome and the RaGoo software version 1.1 (with -b and -C options) (<https://github.com/malonge/RagTag>)22."

3) *Figure 2a. Error margins are missing. Could imprecise estimation (as typically shown with error margins) explain the extreme divergence dates between the cultivars that are suggested by this figure (which are clearly too large)?*

>> Error margins have been added to Figure 2a. It is true that the confidence interval for the divergence between the two *P. armeniaca* cultivars is rather large, but still points to an ancient date [0.92-1.52 Mya], which could be due to very different origins of the two cultivars: Marouch #14 originates from a pool of domesticated apricots introduced thousands years ago from Central Asia

(N_Par or W2 cluster) while cv. Stella's origin is still unknown. Stella's phenotype and phenology are similar to wild *P. armeniaca* trees from Central Asia (small leaves, small fruits, late flowering, high chilling requirement), and is in fact used as a genitor for improving the resistance to pathogens, among which Plum pox virus, but never for its fruit quality or quantity. 'Stella' therefore most likely originates from Central Asian or Chinese forests, from a genetic cluster distinct from the ancestor of European cultivated apricots, Marouch #14 included. This would explain the divergence time between 'Stella' and Marouch #14, as inferred by BEAST (Figure 2a). We have explained this in the manuscript Supplementary note 1.

4) Domestication analysis. Removal of samples could affect the site frequency spectrum. The admixed samples could also result from incomplete lineage sorting. Would this lead to different conclusions? Is it possible to estimate the gene flow indicated in Figure 5c?

>>Indeed, removing individuals affects the site frequency spectrum and this is precisely the point here, as hybrids would bias the site frequency spectrum and thus blur signs of selection. Individuals appearing admixed in barplots result from recent admixture and not from incomplete lineage sorting as Structure analyses identify panmictic groups; polymorphisms present for long within groups, as is the case for incomplete lineage sorting, segregate according to Hardy Weinberg expectations and thus do not generate admixed barplots. Admixture in barplots indicates recent gene flow and can easily be seen precisely in barplots. To estimate more ancient gene flow, we need more sophisticated analyses such as ABC inferences; for detecting ancient gene flow among the panmictic clusters identified in barplots, we need to remove admixed individuals to focus on signals of ancient gene flow. We have tried to clarify this point in the ms.

We estimated gene flow rates for the most likely scenario inferred with ABC-RF (Extended data Table S12), but confidence intervals were pretty large, which is not surprising for ABC inferences. However, coalescent-based ABC inferences are powerful for comparing scenarios, and revealed that, even removing recent footprint of admixture, crop-wild gene flow is still high as depicted in Figure 5. This means that, even when removing recently admixed individuals, we still detect gene flow, having occurred during apricot domestication. We therefore believe that our choice, not only did not bias our conclusions, but was the only way to properly estimate gene flow having occurred during domestication.

5) Would the usage of a Chinese apricot as reference lead to other insights on genes that are under selection in the Chinese cultivar?

>>The microsynteny between the Chinese and European genome assemblies is quite high, so that local selective sweeps detected using the European genome as a reference should also be detected using the Chinese genome assembly. Most genes are shared orthologs between the genomes so the detected genes and functions should also be similar. Obviously, there are some insertions/deletions between genomes and there may be some marginal changes if we used a different genome assembly, as in any population genomic study, but this should not change the main message.

Minor:

1) Minor but important. The font sizes in the figures are consistently too small, and take away the joy of looking at them.

>> The font size of the Figures has been substantially improved and enhanced in the revised version, following reviewer #2's recommendation.

2) *Figure 1d). The amount of structural variation appears too little in number.*

>>In Figure 1d, structural variants are depicted in the tracks C to F. The number of inversions and duplications (E and F tracks) is rather low, as expected for this type of structural variants. However, the number of insertions and deletions (C and D tracks) is much higher, with around 500 of each for 'Stella' and CH320_5 (Extended data Tables S8 and S9). The circos plot depicted in Figure 1d is consistent with the number of structural variations detected with the Bionano Genomics tools. As illustrated in the case of the Manchourian apricot (CH264_4), the structural variations analysis depends on the possibility to align the optical map to the reference (see Supplementary note 6). Indeed, structural variations can only be detected on the genomic fragments where the optical maps can be aligned. Of course though, the number of duplications and translocations detected by the algorithm is probably non-exhaustive. A sentence has been added in the Supplementary information document, Supplementary note 6, to highlight this point.

3) *Line 269-277: numbers do not add up. This could be clearer.*

>> Done. Thank you very much for having detected the mistake.

4) *Line 346: W1 and W2 are used before being defined in line 366.*

>>We have now defined W1 and W2.

5) *Figure 5c: Names at the lower part of the figure are not defined.*

>>We have defined the names below the barplot in the legend.

Reviewer #3 (Remarks to the Author):

Manuscript of "Following the Adaptive Path of Fruit Tree Domestication using Population Genomics: the Case of Apricots" (NCOMMS-21-07239-T) de novo assembled four *Armeniaca* genomes, and re-sequencing 600 apricot accessions to achieve the origin and domestication of apricots. This manuscript provide good materials and large datasets, however, it is not well organized. The main story throughout the MS is not clear. The authors should carefully think about it and reorganize the story. There are some suggestions for this MS.

Line 131. The authors used two sequencing technologies, PacBio and ONT, to sequence the Armeniaca genomes. They should compare their advantages and disadvantages on the genome assembly of perennial fruit trees and this will provide useful information for later researchers.

>>Indeed, we used two different technologies for the reference genomes, but the main focus of the article was not to compare PACBIO and ONT technologies. Additionally, genomes have been sequenced at different times with, now, already fairly old technologies and any comparison would now be outdated information. However, we showed that both long-read technologies can be used to sequence fruit tree genomes.

Line 132,138 What is the appropriate sequencing method for different heterozygosity genome? Could you explain your strategy in detail? Why is it 73X and 60X (PB) for Marouch #14 and Stella, and 113X and 139X (ONT for the others)? Have you tried other fold coverage?

>> A key factor for assembling heterozygous genomes is the read length, and our strategy was therefore to sequence the two highly heterozygous genomes with the longest reads possible (ONT). The coverage is different because the genomes were sequenced at different times. Coverage was thus more the result of the technology power than a sequencing strategy.

Can you give some introduction of the newly assembled accessions of 'Marouch #14', 'Stella', 'CH320-5', and 'CH264.4'?

>>We have added extra informations on those accessions in Supplementary note 1

Figure 1d Why do the authors use Marouch #14 as the reference genome?

>>We selected Marouch as a reference genome because of its homozygosity, thus dealing with one single haplotype for the reads mapping.

*Table 1 'Number of scaffolds *' The unplaced scaffolds could tried to be splited, and then blast to the other references to find the anchored place. If possible, Hi-C data would be an effective way to assist genome assembly.*

>>This could be indeed improved in a future assembly version with new data (such as genetic markers and Hi-C). However, our preliminary annotation of these four scaffolds shows a very high percentage of repeated elements (75% for these four scaffolds versus 37 - 44% in the rest of the genome, see Supplementary note 5). Therefore, we are not sure to what extent anchoring those four scaffolds is worth the effort.

Line 202. An inversion of ca. 600 Kb was detected in the P. armeniaca Marouch #14 genome. This structural variation of 600 Kb should be verified by PCR.

>> The 600Kb inversion was verified by PCR as recommended by reviewer #3. See the new version of Supplementary Figure S10.

Line 250 Re-sequencing depth of all the accessions varies a lot in supp table S1. Is there any SNP calling bias? How could you deal with the data to avoid it? What is more, there are five candidate genomes. The mapping rate also has a big difference. Is it possible to use the pan-genome as the reference to avoid mapping bias?

>>Illumina sequencing yielded quite even sequencing depth, ca 20-21x, so there should be little SNP calling bias arising from sequencing depth. There was indeed a variable mapping rate that may result from genetic distance (for ex. peach, almond or *P. brigantina* reads mapping over the *P. armeniaca* genome), but the mapping rate remained sufficient. Building a pangenome would be beyond the scope of the present study.

Line 249 Have you tried the mitochondrial genome based phylogenetic relationship? How about the nuclear DNA based phylogenetic relationship? Do they have coincidence?

>>Mitochondrial genealogies are often different from nuclear gene genealogies between species, and mitochondrial genealogies do not provide much information on species history, only on the mitochondrial genome history, which was not our goal here. Within species, different nuclear genes have different genealogies, and also different from mitochondrial genealogies, it is not very interesting to study this aspect in our present study.

Line 326 Best K values can be calculated with deltaK.

>>For fastSTRUCTURE (Raj et al 2014), cross validation (i.e. small values of cross-entropy criterion mean better runs) is the recommended method to choose the most relevant number of clusters (K). DeltaK is less powerful, pointing to the strongest population structure and not the finest one, which is usually more biologically relevant.

Line 341 This specification and domestication history would be very interesting, and the authors should be very carefully to verify these results with other strategy, such as genetic diversity methods π , F_{ST} , Tajima's D and IBD etc..

>>The ABC combined with a coalescent-based simulator is a powerful method to infer evolutionary history and is based on the statistics cited by the reviewer (π , F_{ST} , Tajima's D ...). Besides, before running ABC we have explored the population genetic diversity, differentiation and structure, using those different statistics, but we had to use coalescent-based approaches to then infer the apricot domestication history as descriptive statistics were not sufficient. We have added the π value for each population in Figure 5 and the genetic diversity estimates per population in extended data Table S15, together with the former genetic differentiation estimates (F_{ST}).

Line 375. More evidence should be provided for the inference of independent domestication events, for example, identity by descent (IBD) analysis. The author seems to mention this analysis in the method section, however, we have not seen the relevant results to support this inference.

>>Coalescent-based methods provided a strong support for this scenario of independent domestication events. Inference methods are the most powerful and relevant to elucidate domestication scenarios in complex cases as the present one. IBS analyses are mostly descriptive and would not be powerful or relevant here. We did use the IBD approach (Supplementary note 11), but to identify clonemates and first-degree related individuals, in order to avoid biases in allele frequencies.

Have you found any differentiation or group-specific mutations for the variation studies?

>>Estimates of genetic differentiation among populations are provided in extended data Table S15. We have added diversity estimate π for each population in figure 5, and genetic diversity estimates are provided in the new extended data Table S15. D -statistic estimates among the six populations kept for ABC-RF inferences are presented in Supplementary Figure S27, which gives proportions of shared alleles.

Southeast part of China will be an important place for plant origin. Have you consider about the sample collection of this region? Comparisons could be made with other published Rosaceae species's origin and evolution history.

>>We have added in the discussion comparison with the apple, pear and peach domestication histories. We cannot sample further at this stage of the study and anyway germplasm from Southeastern China is irrelevant to the objectives of our study. Southern Eastern China is a region where *P. mume* grows well but *P. armeniaca* does not because of its higher chilling requirement. We have added a sentence in the Supplementary note 1 on this matter. While the Tian Shan-Altay region, particularly the upper Illi Valley in North-Western China, was a refugium for *P. armeniaca* during the last glacial maximum, it was not the case for the South Eastern Chinese region (Meng et al., 2015).

The author has used "enriched" or "significantly enriched" many times, please indicate the significant values. In addition, P_value should be P value (Line 519).

>> Most of the sentences where we used the term 'enriched' or 'significantly enriched' refer to data depicted in extended data Table S22 where we specified at the bottom of each table that the results presented were filtered based on an adjusted p value evaluating the probability that this over-representation of genes belonging to a given class was due to chance. It was then corrected by the multiple test method, with an adjusted p value of 0.05. Adjusted p value and the negative log₁₀ of adjusted p value are displayed in columns D and E of Extended data Table S22. Concerning the candidate genes present in the selective sweep regions, we presented the p value and likelihood estimates in extended data Table S20. Adding those values in brackets to the manuscript would complexify its layout and its reading. We however cited those tables any time we wrote 'enriched' or 'significantly enriched'. P value has been corrected all over the text (MS and supplemental information).

Line 527. The authors identified many genes involved in fruit quality and perennial life cycle traits, but they do not explain their expression levels in the main text. The appropriate expression level verification and related discussion should be contained.

>>We did not conduct differential expression studies for genes under selection. Only part of the new alleles potentially leads to differential expression (the ones in the 5 prime upstream regions of the genes under selection). Some mutations which were fixed by selection are non-synonymous variations within the coding sequence (see results of McDonald Kreitman's test) and would have no effect on the expression of the candidate gene.

In discussion section. The related Rosaceae species domestication process should be discussed.

>>We added general sentences on domestication in perennial fruit trees and we have added more specific statements about what is known in apples, pear and peach.

Line 849. We have not seen the genome data shared by the authors on GDR. The data and annotation information such as gene structure, gene function and TE should be included in it.

>>Data has been submitted in early February at the GDR. They will be released as soon as the paper is accepted. Find below confirmation from the GDR curators.

Sujet : FW: Apricot data

Date : Mon, 8 Feb 2021 14:54:41 +0000

De : Jung, Sook <sook_jung@wsu.edu>

Pour : alexis.gropi@u-bordeaux.fr <alexis.gropi@u-bordeaux.fr>

Copie à : veronique.decroocq@inrae.fr <veronique.decroocq@inrae.fr>, Main, Doreen <dorrie@wsu.edu>

Great thank you! We will wait until you let us know that it's okay to make them public.
Thanks
Sook

From: Sook Jung <sookjc@gmail.com>
Sent: Monday, February 8, 2021 9:40 AM
To: Jung, Sook <sook_jung@wsu.edu>
Subject: Fwd: Apricot data

----- Forwarded message -----

From: Alexis GROUPI <alexis.gropi@u-bordeaux.fr>
Date: Mon, Feb 8, 2021 at 8:20 AM
Subject: Re: Apricot data
To: Sook Jung <sookjc@gmail.com>, dorrie@wsu.edu <dorrie@wsu.edu>
Cc: Veronique Decroocq <veronique.decroocq@inrae.fr>

Dear Sook and Dorie,

Here are all the data from our work on abricots genomics :
<https://filesender.renater.fr/?s=download&token=e9e221d5-5d80-44f7-a4fd-9b334051e269>

The article will be submitted in the next weeks to Nature Genetics :

Following the Adaptive Path of Fruit Tree Domestication using Population Genomics : the Case of Apricots

Alexis Groppi*¹, Shuo Liu*^{2, 3}, Amandine Cornille*⁴, Stéphane Decroocq², Quynh Trang Bui², David Tricon², Corinne Cruaud⁵, Sandrine Arribat⁶, Caroline Belser⁵, William Marande⁶, Jérôme Salse⁷, Cécile Huneau⁷, Nathalie Rodde⁶, William Rhalloussi⁶, Stéphane Cauet⁶, Benjamin Istace⁵, Erwan Denis⁵, Sébastien Carrère⁸, Jean-Marc Audergon⁹, Guillaume Roch^{9, 10}, Patrick Lambert⁹, Tetyana

Zhebentyayeva¹¹, Wei-Sheng Liu³, Olivier Bouchez¹², Céline Lopez-Roques¹², Rémy-Félix Serre¹², Robert Debuchy¹³, Joseph Tran¹⁴, Patrick Wincker⁵, Xilong Chen⁴, Pierre Pétriacq², Aurélien Barre¹, Macha Nikolski¹, Jean-Marc Aury⁵, Albert Glenn Abbott¹⁵, Tatiana Giraud¹⁶, Véronique Decroocq²□

*These authors contributed equally to this work.

□Correspondence should be addressed to V.D. (veronique.decroocq@inrae.fr), T.G. (tatiana.giraud@u-psud.fr) or A.G.A. (albert.abbott@uky.edu)

Keywords: domestication, fruit tree crop, selective sweeps, positive selection, evolution, adaptation, comparative genomics, apricot, QTL

The raw NGS reads have been deposited at the ENA (<https://www.ebi.ac.uk/ena/browser/home>)

Feel free to contact me for any complementary information

Best

Alexis

REVIEWERS' COMMENTS

Reviewer #1 (Remarks to the Author):

The authors have revised the paper according to my suggestions, including adding archaeological context to their timing. The paper is much improved by these changes.

Reviewer #2 (Remarks to the Author):

The authors have addressed all my points in a satisfying way, and I support the publication of the manuscript. I congratulate the authors to the publication of this tour de force.

Reviewer #3 (Remarks to the Author):

Southwest part of China will be an important place for plant origin. Correction for "southwest".

The authors have addressed my questions.